# DSENet: A Novel Dual-Stream Enhancement Network for Multi-Scale Non-Stationary Time Series Forecasting

**Yuhan Wang** [1]  **Yuanyuan Zou** [1]  **Jie Cheng** [1]  **Bin Dai** [1]  **Jinhong Guo** [1 2]

## Abstract

Accurately capturing local variations in long series has always been one of the most challenging problems in time-series forecasting especially in medical signals, where local variations often indicate pathological events. Our study reveals a previously overlooked key bottleneck in this field: traditional global and local branches learn similar representations, leading to strong feature coupling and reduced sensitivity to local variations. To address this challenge, we propose the novel Dual-Stream Enhancement Mechanism, which structurally enlarges the difference between global and local patterns, enabling weak interactions between the two. Based on this idea, we introduce a new baseline model for blood glucose prediction: Dual-Stream Enhancement Network (DSENet), which fundamentally alleviates the problem of excessively strong coupling between global and local features. Experimental results show that our model achieves SOTA performance on multiple public datasets. Moreover, benefiting from extremely low computational cost, our model demonstrates strong application potential and can serve as a baseline model in multiple domains in the future. The code is avaliable at: https://github.com/yhwang303/DSENet

## 1. Introduction

Time-series forecasting is crucial in important applications such as climate monitoring, financial analysis, and medi-

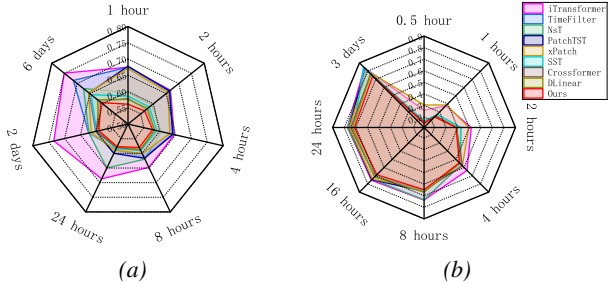

*Figure 1.* (a) Comparision MAE results of DCLP5 dataset with different time-scales. (b) Comparision MAE results of zero-shot prediction results of OhioT1DM dataset with differnet time-scales

cal alerting. Advances in deep learning have significantly enhanced the capability of models to characterize complex dependencies and multi-scale dynamics, especially promoting the development of long-term time-series forecasting, (Dama et al., 2025) while maintaining lightweight properties (Lin et al., 2024). However, real-world series commonly exhibit strong non-stationarity and abrupt fluctuations, and existing methods remain insufficient in modeling local drastic changes. Physiological signals are typical strongly non-stationary time series. Modeling local variations in complex physiological time series and accurately predicting future variations has long been a persistent challenge in the medical field. Blood glucose trajectories are particularly complex. Although personalized modeling (Odnoblyudova et al., 2023; Brügger et al., 2025) and interpretable abrupt-change prediction methods (Duckworth et al., 2024) have made some progress, their sensitivity to local sudden changes remains insufficient and their generalization ability is limited. Second, another challenging is that when the input sequences become longer, it becomes even more difficult for models to capture local abrupt changes from long time-series, where global trends dominate the prediction process and subtle local variations are easily smoothed out or absorbed by global representations. Previous attempts such as long-short-term separation modeling (Ding et al., 2025) or dual-branch architectures (Zhao et al., 2023) can improve long-range dependency modeling and local perception to some extent. Nevertheless, as the series length increases, their performance in capturing local abrupt changes still degrades significantly.

[1]School of Automation and Intelligent Sensing, Shanghai Jiao Tong University, 800 Dongchuan Road, Shanghai, 200240, China. [2]Research Center of Internet of Medical Things, Qilu Hospital of Shandong University, Jinan, Shandong, 250012, China.. Correspondence to: Yuanyuan Zou <zoey_029@sjtu.edu.cn>, Bin Dai <daibin@sjtu.edu.cn>, Jinhong Guo <guojinhong@sjtu.edu.cn>.

*Proceedings of the $43^{rd}$ International Conference on Machine Learning*, Seoul, South Korea. PMLR 306, 2026. Copyright 2026 by the author(s).

To address the aforementioned challenges, we propose a novel dual-stream model in the field of blood glucose prediction, where the global stream is dedicated to modeling the overall temporal trends of the sequence, while the local stream focuses on capturing local abrupt variations. Meanwhile, we argue that the degraded ability of existing models to detect local fluctuations in long-horizon forecasting primarily stems from the strong coupling between global and local features in high-dimensional representations (Wang et al., 2023). Their feature differences are small, resulting in mutual interference. To address this issue, we introduce a dual-stream enhancement mechanism that significantly enlarges the representational discrepancy between global and local patterns at the architectural level. Specifically, we enhance the global stream using our proposed Dynamic Fusion Bidirectional Mamba(DFB-Mamba) model to strengthen its modeling capability in long-term series. This structure allow it to better focus on global trends while suppressing local disturbances and can be regarded as a low-pass filter. For the local stream, we design a Low-Rank Enhancement (LoRE) module to extract low-rank local features more effectively, thereby improving the sensitivity to abrupt changes and preventing local information from being overshadowed by global features, it is more like a high-pass filter. By enhancing the two streams separately and widening the gap between global and local patterns, we demonstrate that clearly separating the two paradigms enables the model to improve both trend prediction and abrupt-change detection simultaneously, which is a point that previous work has never achieved. Our model achieves near *SOTA* performance on two publicly available datasets in the blood glucose forecasting field Figure 1.

We term the proposed model as Dual-Stream Enhancement Network (DSENet), mainly to emphasize the network uses two parallel streams after enhancement to model global pattern and local pattern separately. The main contributions of this paper are summarized as follows:

- We follow the dual-stream network design and divide the model into a global stream and a local stream, which capture long-range dependencies and local fluctuations separately. The outputs of the two streams are dynamically fused by the DSSAF algorithm.

- We further introduce the Dual-Stream Enhancement Mechanism to minimize the interference between global and local patterns. To the best of our knowledge, we are the first to explicitly enlarge the discrepancy between global and local paradigms.

- We establish a new baseline model in the field of blood glucose prediction, which achieves accurate multi-scale prediction while maintaining linear computational complexity and extremely low memory usage.

## 2. Related Work

### 2.1. Dual-Stream Architecture

Dual-stream models (Simonyan & Zisserman, 2014) have emerged as an effective paradigm for feature decomposition and collaborative learning, showing great potential in both visual and sequence understanding tasks. This architecture was first introduced in video understanding, and later achieved rapid progress in image and speech processing (Choi et al., 2023; Yu et al., 2024; Shen et al., 2025). In the time-series domain, Stitsyuk & Choi (2025) proposed the dual-branch architecture, where the two branches capture seasonal information and trend separately. Although dual-stream architectures explicitly separate global and local branches, a key bottleneck remains that such designs do not necessarily yield disentangled representations (Tonekaboni et al., 2022). Extensive or unconstrained cross-stream interactions introduce feature coupling and redundancy, which blurs the functional boundaries between streams (Wang et al., 2025; Yi & LianLei, 2025).

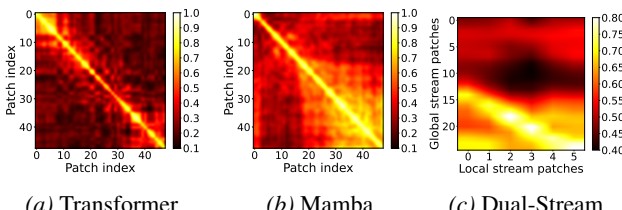

*(a) Transformer*   *(b) Mamba*   *(c) Dual-Stream*

*Figure 2.* Centered Kernel Alignment (CKA) Visulization results show Mamba performs better in global pattern while Transformer performs better in modeling local variations.

### 2.2. Blood Glucose Prediction

Blood glucose dynamics exhibit complex nonlinear patterns influenced by various physiological, behavioral, and environmental factors. In 2014, Zecchin et al. (2014) proposed using jump neural networks to forecast blood glucose levels 30 minutes ahead. Recurrent models such as RNN and LSTM (Sherstinsky, 2020) have also been widely applied in blood glucose forecasting (Mohebbi et al., 2020; Wang et al., 2020; Alshehri et al., 2024; Mazgouti et al., 2025). With the emergence of Transformers (Vaswani et al., 2017), an increasing number of sequence-to-sequence models have been utilized in this (Lee et al., 2023; Sergazinov et al., 2023; Karagoz et al., 2025). However, most existing Transformer-based methods still rely on a single or weakly coupled attention scale, limiting their ability to adapt to heterogeneous physiological scenarios. What's more, to reduce computational complexity, state-space-model-based Mamba architectures have also been introduced for blood glucose forecasting (Şentürk et al., 2025). Moreover, with the rapid development of large language models (LLMs), LLM-based approaches have been explored for this task (Li

et al., 2025), achieving promising results. However, since these models typically rely on a single modeling paradigm for glucose time series, they generally lack sensitivity to abnormal changes, which are closely associated with clinically hazardous events such as postprandial hyperglycemia and exercise-induced hypoglycemia.

## 3. Preliminaries

### 3.1. Problem Statement

In time series forecasting, the model receives input as a look-back window $\mathbf{L} = (x_1, x_2, \ldots, x_L) \in \mathbb{R}^{L \times M}$, and equivalently $\mathbf{L} = \left(\mathbf{x}^{(1)}, \mathbf{x}^{(2)}, \ldots, \mathbf{x}^{(M)}\right) \in \mathbb{R}^{L \times M}$, with length $L$ and $M$ variates. Separately, each of them can be viewed in temporal domain $\mathbf{x}_t \in \mathbb{R}^{1 \times M}$ or in the variate dimension $\mathbf{x}^{(m)} \in \mathbb{R}^{L \times 1}$. The goal is to forecast future $F$ values $\mathbf{L}_{\text{out}} = (x_{L+1}, x_{L+2}, \ldots, x_{L+F}) \in \mathbb{R}^{F \times M}$.

### 3.2. Motivation of Using Dual-Stream Architecture

Conventional time-series models, such as DLinear (Zeng et al., 2023) and Transformer-based approaches(Wu et al., 2021; Sasal et al., 2022; Wang et al., 2023), typically adopt a single-branch architecture. However, such designs generally fail to simultaneously achieve robust modeling of global trends and precise sensitivity to local variations. To address this limitation, we argue that a dual-stream architecture with explicitly decoupled modeling responsibilities is necessary.

As shown in Figure 2a and Figure 2b, Mamba performs better in global pattern while Transformer performs better in local variations, there is complementarity between them. However, a simple dual-stream design still faces two limitations: (1) Traditional fusion methods often concatenate the two paradigms along the feature dimension (Ye et al., 2025), ignoring the structural properties of the original sequence. (2) The boundary between global patterns and local patterns is inherently ambiguous. As shown in Figure 2c, the dual-stream architecture composed of Vanilla Transformer and Vanilla Mamba exhibits significant feature coupling. This results in severe mutual interference persisting during feature modeling for dual streams. Thus, an effective dual-stream framework must not only structurally separate the two modes, but also enlarge their representational differences and weaken their interactions in order to truly improve overall performance.

## 4. Methodology

Overall, as illustrated in Figure 3, our DSENet mainly consists of a dual-stream structure along with two embedded algorithms. The DFB-Mamba block is responsible for the global stream, modeling the global patterns of the sequence. The LoRE block cooperates with the Light-Weight SA block

---

**Algorithm 1** DMSP

**Input:** Glucose Time Series $x \in \mathbb{R}^{M \times L}$.
**Output:** Results after patching: $X_g \in \mathbb{R}^{M \times N_g \times P_g}$ for global patching; $X_l \in \mathbb{R}^{M \times N_l \times P_l}$ for local patching.
**if** $F \leq F^*$ **then**
    $(P_g, S_g) \leftarrow (P_g^{(0)}, S_g^{(0)})$
    $\beta \leftarrow \max(\beta_{\min}, F/F^*)$
    $(P_l, S_l) \leftarrow \big(\text{clip}(\text{round}(\varphi_\theta(F; P_g^{(0)})), P_{\min}, P_{\max}),$
    $\text{clip}(\text{round}(\varphi_\theta(F; S_g^{(0)})), S_{\min}, S_{\max})\big)$
**else**
    $(P_l, S_l) \leftarrow (P_l^{(0)}, S_l^{(0)})$
    $\beta \leftarrow \min(\beta_{\max}, F/F^*)$
    $(P_g, S_g) \leftarrow \big(\text{clip}(\text{round}(\varphi_\theta(F; P_l^{(0)})), P_{\min}, P_{\max}),$
    $\text{clip}(\text{round}(\varphi_\theta(F; S_l^{(0)})), S_{\min}, S_{\max})\big)$
**end if**
**Enforce** $L > [N_i = \lfloor (L - P_i)/S_i \rfloor + 1] > 1, i \in \{g, l\}$
$X_g \leftarrow \text{Patch}(x, P_g, S_g),\ X_l \leftarrow \text{Patch}(x, P_l, S_l)$
**return** $X_g, X_l$

---

to form the local stream, enabling the perception of local abrupt changes. The Dynamic Multi-Scale Patching Algorithm (DMSP) divides the sequence into coarse-grained global sequences and fine-grained local fluctuations under different granularities, and outputs a patching structure that can be processed by the dual streams. The Dual-Stream State Adaptive Fusion (DSSAF) algorithm fully considers the characteristics of the original sequence and generates two stream-related weights to guide the final fusion.

### 4.1. Dynamic Multi-Scale Patching Algorithm(DMSP)

**Patching**. The model receives $\mathbf{L} = (x_1, x_2, \ldots, x_L) \in \mathbb{R}^{L \times M}$ as input and performs patching with different granularities in two streams. The output of global patching is denoted as $\mathbf{X}_g \in \mathbb{R}^{M \times N_g \times P_g}$ and the output of local patching is denoted as $\mathbf{X}_l \in \mathbb{R}^{M \times N_l \times P_l}$, where $N_i = \lfloor (L - P_i)/S_i \rfloor + 1, \quad i \in \{g, l\}$ represents the number of tokens after patching. $P$ represents the patch length and $S$ denotes the stride. For different $F$, we adopt different patching strategies. Specifically, when $F$ is small, we fix $P_g$ and $S_g$, only adjust $P_l$ and $S_l$. When $F$ is large, we fix $P_l$ and $S_l$, only adjust $P_g$ and $S_g$. This enables the model to apply different patching granularities for different temporal scales, thereby enhancing its multi-scale representation capability.

**Dynamic Temporal Granularity**. As shown in Figure 2, the global pattern and the local pattern behave differently when modeling the series, since they focus on opposite aspects. Qualitatively, for the global stream, the patching granularity should be coarse so that the model concentrates on extracting global features. For the local stream, the patching granularity should be fine so that the model focuses on

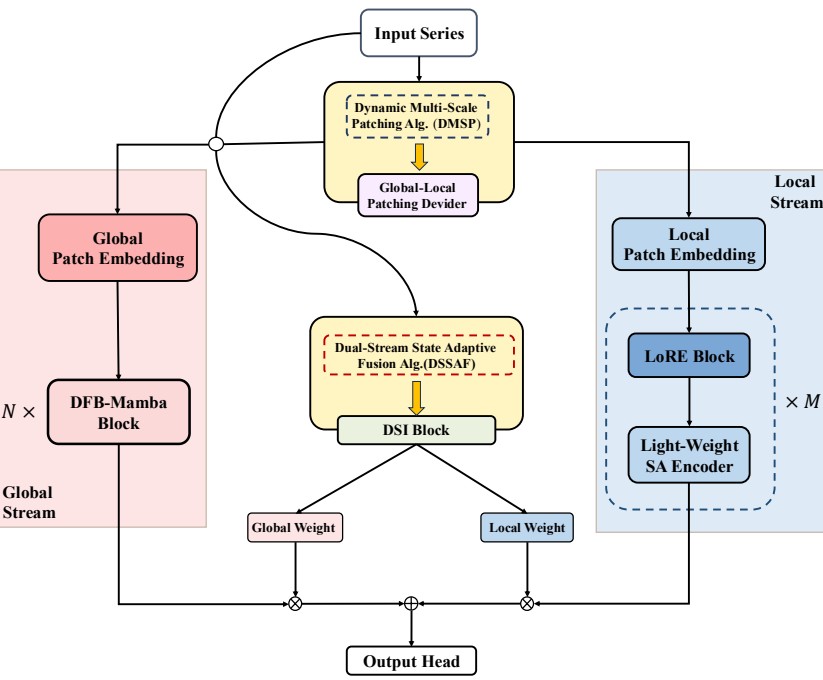

*Figure 3.* The overall structure of the DSENet. It has two enhancement streams: global stream and local stream. DSSAF Alg. combines the two streams under the guide of two dynamic weights generated from the original input series.

sensing local abrupt changes. To quantitatively analyze the notion of granularity, we introduce a new concept, Dynamic Temporal Granularity, denoted as $\zeta$ which guides the generation of patches. Different from resolution (Zhao et al., 2023; Xu et al., 2024), we consider granularity to be closely related to the forecast length. We first define a scaling factor that is associated with the forecast length as follows:

**Definition 4.1.** The scaling factor for the forecast length denoted as $\beta(F)$, the formula is: $\beta(F) = \text{clip}(F/F^*, \beta_{\min}, \beta_{\max})$

$F^*$ here is the threshold (Default=12) to distinguish between short-term and long-term predictions, while specifying empirical upper and lower bound $\beta_{\max}$ and $\beta_{\min}$. Definition 4.1 ensures that as the prediction length $F$ increases, $\beta(F)$ remains non-decreasing, thereby enabling a smooth transition between patch length and stride for short and long series. Based on this, we further derive expressions for the patch length and stride by using two hyper-parameters $\alpha_p$ and $\alpha_s$:

$$\varphi_\theta(P_i) = \alpha_p \cdot P^{(0)} \cdot \beta(F), \quad \varphi_\theta(S_i) = \alpha_s \cdot S^{(0)} \cdot \beta(F) \quad (1)$$

Since $\beta(F)$ is non-decreasing, Equation (1) explicitly shows that as the prediction length increases, both $P_i$ and $S_i$ expand synchronously, leading to an enlargement of the receptive field. Conversely, when the forecasting length decreases, the patch parameters shrink, resulting in a denser patch distribution that emphasizes local mutations.

**Theorem 4.2.** $\zeta$ *can be described only related to $F$:*
$\zeta_i(F) \approx \frac{1}{S_i \sqrt{P_i}} = c \cdot \beta(F)^{-3/2}$

where $c$ is constant that combines $P^{(0)}$, $S^{(0)}$, $\alpha_p$ and $\alpha_s$. Theorem 4.2 gives the expression of Dynamic Temporal Granularity $\zeta$. The detailed proof will be shown in the **Appendix** A. It can be observed that the granularity $\zeta$ is a monotonically decreasing function with respect to $F$ and it can be fitted in two ways: a joint strategy that adjusts both the patch length and the stride, or a strategy that directly fits granularity using only $F$. The complete DMSP algorithm is provided in Algorithm 1. DMSP performs patching on the global pattern and the local pattern using two different patching strategies determined by the magnitude of the forecast length $F$. The dynamic temporal granularity corresponding to the given forecast length $F$ is first computed, and it is then used to guide the selection of $P_i$ and $S_i$. This allows the model to adjust the patch length and stride for the global pattern and the local pattern in a principled manner, achieving optimal forecasting performance.

### 4.2. Dual-Stream Architecture

**Global Stream**. Mamba demonstrates strong capability in capturing global features with linear computational complexity, as shown in Figure 2. Therefore, we adopt Mamba as the global stream. To further enlarge the difference between the global pattern and the local pattern, we enhance the global stream to strengthen its ability to model global features and reduce the influence of local fluctuations. We propose a new model: ***DFB-Mamba***. We introduce a forget gate (Liang et al., 2024) to dynamically regulate the combi-

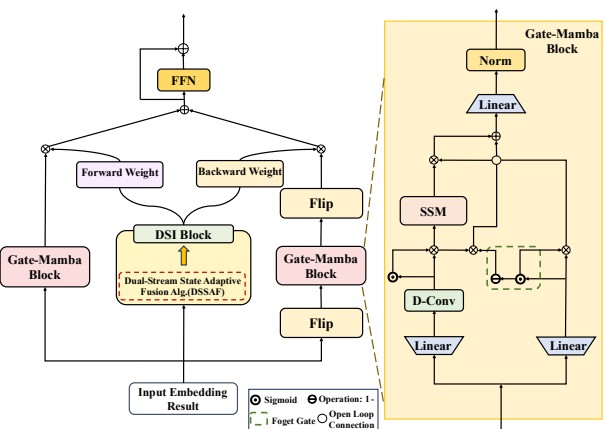

*Figure 4.* The overall structure of proposed DFB-Mamba, which is performed as the enhancement global stream. Gate-Mamba is equipped with the forget gate and dynamic convolution.

nation of the two branches in Mamba, improving its long-range modeling capability, as illustrated in Figure 4. The input series has dimension $(M, N_g, P_g)$. The two branches are defined as left branch $x \in \mathbb{R}^{M \times N_g \times P_g}$ and right branch $z \in \mathbb{R}^{M \times N_g \times P_g}$. The right branch is projected through a linear layer and then passed through the forget gate, where its output is multiplied with the after $SiLU$-activated left branch $x'$, which produces one of the outputs. The other output comes from the selective state update of the SSM in the left branch. These two outputs are combined as the output of the Gate-Mamba block. In addition, we replace the convolution layer in the original Mamba with a learnable dynamic convolution layer to adapt to inputs with different temporal scales. The kernel size is adjusted according to the input sequence length, enabling more effective pre-processing. Finally, the output of Gate-Mamba $y \in \mathbb{R}^{M \times N_g \times P_g}$ is computed by the following:

$$y = \text{SSM}(x') \otimes SiLU(z) + x' \otimes (1 - \sigma(z)) \quad (2)$$

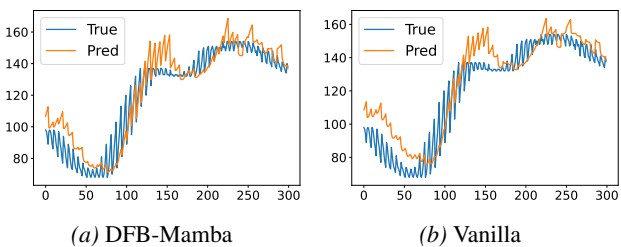

*(a)* DFB-Mamba   *(b)* Vanilla

*Figure 5.* Forecasting results show that our proposed DFB-Mamba performs better than Vanilla Mamba in modeling global pattern.

In addition, to further enhance the representational capacity of sequential features, we introduce a bidirectional modeling strategy. By leveraging both forward and backward

temporal dependencies, the model gains improved capability in capturing global patterns. For the fusion of the bidirectional sequences, we employ a Dynamic State Interaction (DSI) block embedded with the DSSAF algorithm. Figure 5 compares our proposed DFB-Mamba model with the original Mamba model on a blood glucose forecasting example, demonstrating that our model achieves stronger global modeling capability.

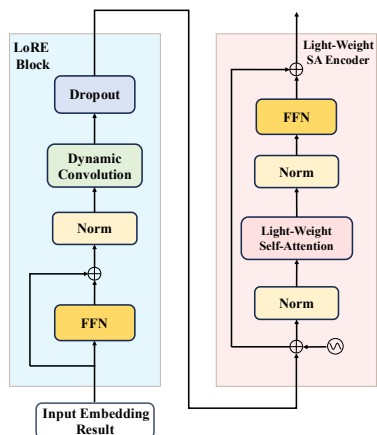

*Figure 6.* The overall structure of proposed LoRE block, followed by the Light-Weight Self-Attention encoder.

**Local Stream**. As shown in Figure 2, Transformer-based models demonstrate strong advantages in capturing local fluctuations. However, due to the quadratic computational complexity of the self-attention mechanism, modeling local variations over long sequences requires substantial computational resources. To address this issue, we propose a lightweight self-attention module Figure 6 that adopts a local windowing strategy (Parmar et al., 2018), restricting the computation of attention scores to within each window. The specific computation is expressed as follows:

$$\text{Attention}(Q_l, K_l, V_l) = \text{softmax}\left(\frac{Q_l K_l^\top}{\sqrt{d_l}}\right) V_l \quad (3)$$

where $Q_l$, $K_l$, $V_l$ denotes query, key, value within local window respectively. What's more, the window forces the model to focus only on the relationships among tokens within the local region while ignoring tokens that are far from the current window, which can significantly enhance its sensitivity to local variations.

Similarly, the local stream also requires further enhancement in capturing local features, thereby enlarging its difference from the global pattern. To this end, we propose a new module called **LoRE**. The goal of LoRE is to extract stable low-rank patterns within each temporal patch while preserving high-frequency local dynamics. It takes as input the embedded local patching results, denoted as $X \in \mathbb{R}^{M \times N_l \times P_l}$. We

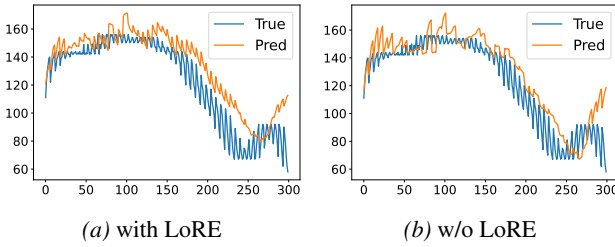

*(a) with LoRE*    *(b) w/o LoRE*

*Figure 7.* Forecasting results indicate our proposed LoRE block can significantly enhance the ability to capture local variations.

first apply a feed-forward network (FFN) to learn the low-rank structural features of the input. Specifically, two projection matrices $W_1 \in \mathbb{R}^{m \times k}$, $W_2 \in \mathbb{R}^{k \times m}$ are used to compress the high-dimensional vector into a low-dimensional subspace and then project it back to the high-dimensional space. This operation performs a low-rank linear transformation within each local patch and preserves its principal directions. The dynamic convolution in LoRE also aggregates convolution kernels adaptively based on the input, allowing the representational capacity of the module to change dynamically. This local dynamic modeling further emphasizes small-scale variations within each patch.

**Theorem 4.3.** *We denote $u \in \mathbb{R}^P$ as the corresponding patch vector, the LoRE block applies $\mathbf{u} \xrightarrow{f} \tilde{\mathbf{u}} = \text{LN}(\mathbf{u} + \text{FFN}(\mathbf{u})) \xrightarrow{dynamic\ conv} \hat{\mathbf{u}} = \mathcal{T}(w(\tilde{\mathbf{u}}))\,\tilde{\mathbf{u}}$. By performing linearization around reference point $\tilde{\mathbf{u}}$, the Jacobian Matrix of LoRE can be obtained as: $J_{\text{LoRE}}(\tilde{\mathbf{u}}) \approx \mathcal{T}(w(\tilde{\mathbf{u}}))\,J_f(\tilde{\mathbf{u}}) + terms\ depending\ on\ \nabla w(\tilde{\mathbf{u}})$.*

Theorem 4.3 identifies the Jacobian matrix representation of the LoRE block, jointly determined by the FFN output and the convolution kernel $W$, demonstrating that the LoRE block enhances the model's sensitivity to local fluctuations. The detailed proof is provided in the **Appendix** B. Figure 7 illustrates the enhancement effect of the proposed LoRE block on local perception. It can be clearly observed that the attention-based model equipped with the LoRE block is able to model local abrupt changes more effectively.

### 4.3. Dual-Stream State Adaptive Fusion(DSSAF)

**DSI Block**. Existing dual-stream models typically fuse the outputs of the two streams by directly concatenating them along the feature dimension. However, this strategy implicitly assumes that the two patterns are equally important. However, the sequence characteristics of different samples exhibit significant differences: For blood glucose sequences from stable or healthy individuals, the overall trend should dominate; whereas in data from diabetic patients with frequent fluctuations, local patterns hold greater discriminative value. Simple concatenation fails to adaptively adjust the contributions of the two heterogeneous streams, which may

---

**Algorithm 2** DSSAF

**Input:** Original series $\mathbf{x} \in \mathbb{R}^{L \times M}$; Two streams $\mathbf{S}^{(1)}, \mathbf{S}^{(2)} \in \mathbb{R}^{N_1 \times M}, \mathbb{R}^{N_2 \times M}$
**Output:** Fusion result $\mathbf{S}^{(f)} \in \mathbb{R}^{N \times M}$
$\boldsymbol{\mu} \leftarrow \text{MeanPool}(\mathbf{x})$
$\mathbf{z}_0 \leftarrow \text{LayerNorm}(\boldsymbol{\mu})$
$\mathbf{g} \leftarrow \sigma(W_g \mathbf{z}_0 + \mathbf{b}_g) \in \mathbb{R}^{d_g}$
$\mathbf{z}_{\text{cat}} \leftarrow \text{Concat}(\mathbf{z}_0, \mathbf{g}) \in \mathbb{R}^{d_g + d}$
$\tilde{\mathbf{z}} \leftarrow \text{GELU}(W_z \mathbf{z}_0 \oplus \mathbf{g} + \mathbf{b}_z) \in \mathbb{R}^{d_z}$
$\mathbf{h} \leftarrow \text{GELU}(W_1 \tilde{\mathbf{z}} + \mathbf{b}_1) \in \mathbb{R}^{d_h}$
$\boldsymbol{\alpha} \leftarrow W_0 \mathbf{z}_{\text{cat}} + W_2 \mathbf{h} + \mathbf{b}_2 \in \mathbb{R}^2$
$\mathbf{w} \leftarrow \text{Softmax}(\boldsymbol{\alpha}/\tau)$
$w_1 \leftarrow \mathbf{w}[0], \quad w_2 \leftarrow \mathbf{w}[1]$
$\mathbf{S}^{(f)} \leftarrow w_1 Proj(\mathbf{S}^{(1)}) + w_2 \mathbf{S}^{(2)}$
**return** $\mathbf{S}^{(f)}$

---

dilute critical information. To address this issue, we design a dynamic and adaptive fusion algorithm, termed DSSAF as shown in Algorithm 2, which is embedded into the DSI block. It takes the original sequence as input and outputs two weights to enable dynamic fusion of the two streams.

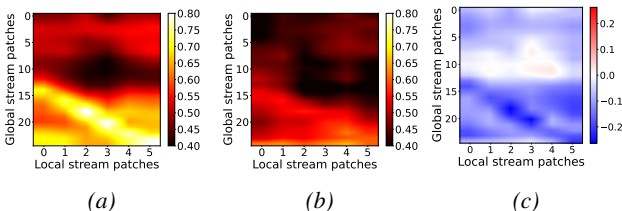

*(a)*    *(b)*    *(c)*

*Figure 8.* The visualization results of CKA approach. (a) The architecture without Dual-Stream Enhancement Mechanism. (b) The DSENet. The degree of feature coupling between the two streams has decreased significantly. (c) The difference CKA figure between (a) and (b), which indicates the Dual-Stream Enhancement Mechanism enlarges the difference between the two streams.

### 4.4. Analysis of Dual-Stream Enhancement Mechanism

**Motivation**. We argue that the primary reason for the suboptimal performance of existing dual-stream time-series forecasting models lies in the mutual interference between the two streams, where a large amount of feature sharing occurs. Therefore, we address this issue from a structural perspective by explicitly encouraging the two streams to learn more distinctive information and clearly differentiate the features each stream focuses on, which constitutes our proposed dual-stream enhancement mechanism. From a theoretical analysis perspective, we demonstrate that this mechanism enables the model to learn features with significantly improved discriminability. Detailed analysis is provided in the **Appendix** C.

**Effectiveness**. We use CKA to further visualize the effectiveness and novelty of the proposed dual-stream enhance-

*Table 1.* Blood Glucose Prediction results with forecasting lengths $FL = \{0.5h, 1h, 2h, 4h, 8h, 16h, 24h\}$ and historical lengths $HL = \{1h, 2h, 4h, 8h, 24h, 2days, 6days\}$. Results are averaged from all forecasting lengths. The best is in **bold** and second-best is underlined. Full results are listed in Table 4 and Table 5 in **Appendix D**.

| OhioT1DM | **Ours** | | TimeFilter | | xPatch | | SST | | DLinear | | iTransformer | | Crossformer | | PatchTST | | NsT | |
|---|---|---|---|---|---|---|---|---|---|---|---|---|---|---|---|---|---|---|
| | MSE | MAE | MSE | MAE | MSE | MAE | MSE | MAE | MSE | MAE | MSE | MAE | MSE | MAE | MSE | MAE | MSE | MAE |
| HL = 1 hour | **0.465** | **0.477** | 0.720 | 0.555 | 0.715 | 0.552 | 0.473 | 0.483 | 0.478 | 0.491 | 0.724 | 0.557 | 0.475 | 0.486 | 0.713 | 0.554 | 0.539 | 0.513 |
| HL = 2 hours | **0.468** | **0.481** | 0.679 | 0.544 | 0.681 | 0.543 | 0.476 | 0.485 | 0.474 | 0.486 | 0.698 | 0.553 | 0.477 | 0.492 | 0.671 | 0.544 | 0.513 | 0.487 |
| HL = 4 hours | **0.466** | **0.478** | 0.611 | 0.523 | 0.614 | 0.522 | 0.478 | 0.494 | 0.476 | 0.488 | 0.639 | 0.540 | 0.468 | 0.483 | 0.605 | 0.527 | 0.518 | 0.492 |
| HL = 8 hours | 0.478 | **0.482** | 0.553 | 0.506 | 0.564 | 0.508 | **0.473** | 0.488 | 0.479 | 0.490 | 0.610 | 0.539 | 0.476 | 0.487 | 0.546 | 0.508 | 0.531 | 0.499 |
| HL = 24 hours | **0.474** | **0.482** | 0.510 | 0.499 | 0.546 | 0.512 | 0.482 | 0.490 | 0.479 | 0.490 | 0.639 | 0.571 | 0.487 | 0.493 | 0.515 | 0.513 | 0.577 | 0.535 |
| HL = 2 days | **0.484** | **0.492** | 0.533 | 0.509 | 0.581 | 0.528 | **0.484** | 0.495 | 0.492 | 0.496 | 0.691 | 0.600 | 0.503 | 0.500 | 0.537 | 0.518 | 0.574 | 0.529 |
| HL = 6 days | 0.488 | 0.498 | 0.537 | 0.510 | 0.543 | 0.527 | **0.487** | 0.503 | 0.488 | **0.496** | 0.635 | 0.601 | 0.514 | 0.510 | 0.587 | 0.532 | 0.645 | 0.563 |

| DCLP5 | **Ours** | | TimeFilter | | xPatch | | SST | | DLinear | | iTransformer | | Crossformer | | PatchTST | | NsT | |
|---|---|---|---|---|---|---|---|---|---|---|---|---|---|---|---|---|---|---|
| | MSE | MAE | MSE | MAE | MSE | MAE | MSE | MAE | MSE | MAE | MSE | MAE | MSE | MAE | MSE | MAE | MSE | MAE |
| HL = 1 hour | **0.609** | **0.561** | 0.949 | 0.675 | 0.946 | 0.671 | 0.632 | 0.586 | 0.627 | 0.576 | 0.953 | 0.675 | 0.617 | 0.578 | 0.954 | 0.678 | 0.705 | 0.598 |
| HL = 2 hours | **0.616** | **0.565** | 0.904 | 0.665 | 0.899 | 0.657 | 0.634 | 0.584 | 0.633 | 0.577 | 0.915 | 0.667 | 0.624 | 0.572 | 0.903 | 0.663 | 0.694 | 0.594 |
| HL = 4 hours | **0.633** | **0.578** | 0.822 | 0.643 | 0.811 | 0.632 | 0.651 | 0.591 | 0.641 | 0.584 | 0.832 | 0.646 | 0.641 | 0.584 | 0.817 | 0.641 | 0.685 | 0.605 |
| HL = 8 hours | **0.630** | **0.580** | 0.731 | 0.613 | 0.720 | 0.601 | 0.644 | 0.590 | 0.634 | 0.587 | 0.800 | 0.648 | 0.641 | 0.588 | 0.727 | 0.617 | 0.711 | 0.620 |
| HL = 24 hours | **0.618** | **0.577** | 0.656 | 0.587 | 0.678 | 0.589 | 0.626 | 0.584 | 0.618 | 0.578 | 0.852 | 0.687 | 0.624 | 0.580 | 0.685 | 0.600 | 0.799 | 0.650 |
| HL = 2 days | **0.650** | **0.592** | 0.745 | 0.624 | 0.728 | 0.616 | 0.663 | 0.599 | 0.660 | **0.592** | 0.958 | 0.734 | 0.677 | 0.600 | 0.688 | 0.602 | 0.730 | 0.623 |
| HL = 6 days | **0.669** | **0.606** | 0.968 | 0.715 | 0.789 | 0.655 | 0.729 | 0.645 | 0.690 | 0.620 | 0.938 | 0.751 | 0.676 | 0.612 | 0.790 | 0.653 | 0.829 | 0.675 |

ment mechanism, as shown in Figure 8. For the original model without the dual-stream enhancement mechanism, strong feature coupling between the global and local streams can be observed in Figure 8a. In contrast, after the two streams are enhanced separately, the feature correlation between the two streams is substantially reduced, as shown in Figure 8b. This observation suggests the proposed design effectively promotes feature decoupling between the two streams. Furthermore, the CKA difference map in Figure 8c provides more direct evidence that DSENet enlarges the representational discrepancy between the two streams.

### 4.5. Analysis of Computational Complexity

Overall, our model maintains linear computational complexity with respect to the input series length $L$. The global stream is built upon Mamba and has a computational complexity of $\mathcal{O}(L)$. The local stream adopts a lightweight attention mechanism, whose per-layer complexity is $\mathcal{O}(N_l/w \cdot w^2 d) = \mathcal{O}(N_l w d) = \mathcal{O}(L)$, where $w$ is the size of window, $d$ is the dimension of model. As a result, the model preserves linear computational complexity across multiple time scales.

## 5. Experiments

### 5.1. Experimental Setup

**Dataset.** To comprehensively evaluate the performance of our proposed model on non-stationary time-series fore-casting, we conduct experiments on two public datasets in the blood glucose forecasting domain: OhioT1DM and DCLP5. The OhioT1DM dataset consists of an 8-week clinical trial involving 12 patients with type 1 diabetes (Marling & Bunescu, 2020). The DCLP5 dataset includes adolescents and adults with T1D who participated in a 6-month randomized multicenter trial. We also verified the generalization ability of our DSENet in several real-world public datasets(**Appendix H**).

**Baselines and Metrics.** We compare our model with several state-of-the-art approaches from the blood glucose prediction domain, including PatchTST (Nie, 2022), Nonstationary Transformer (NsT, (Liu et al., 2022)), iTransformer (Liu et al., 2023), Crossformer (Zhang & Yan, 2023), DLinear (Zeng et al., 2023), SST (Xu et al., 2024), xPatch (Stitsyuk & Choi, 2025) and TimeFilter (Hu et al., 2025). We use the same parameters set in these models and consistently adopt MSE and MAE to measure the predictive performance of models. In addition, we employ the Clarke Error Grid (CEG) to assess clinical risk, providing a more comprehensive evaluation framework.

**Experimental Setting.** To enable multi-scale prediction while balancing clinical relevance and general applicability, corresponding to short-term variations warnings, mid-term intervention decisions and long-term trend assessment. Specifically, the historical sequence length is set to $L = \{1h, 2h, 4h, 8h, 24h, 2days, 6days\}$ and the corresponding forecast horizon is $F = \{0.5h, 1h, 2h, 4h, 8h, 16h, 24h\}$. For each $(L, F)$ combination, we assign different patch

lengths and strides for the global stream and the local stream to match the corresponding temporal scale through DMSP.

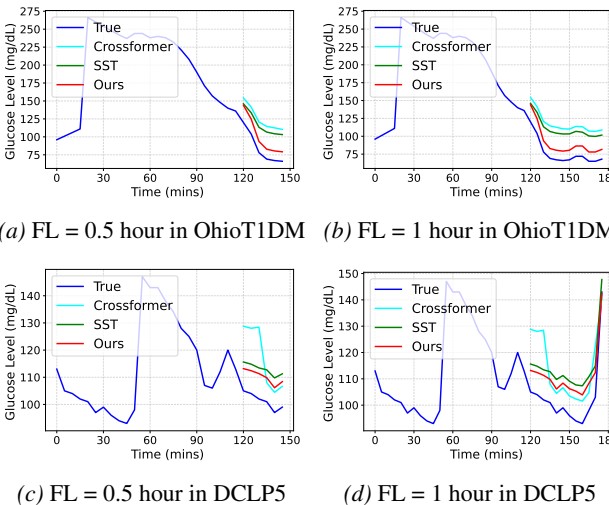

*(a)* FL = 0.5 hour in OhioT1DM    *(b)* FL = 1 hour in OhioT1DM

*(c)* FL = 0.5 hour in DCLP5    *(d)* FL = 1 hour in DCLP5

*Figure 9.* Comparision results in two datasets show that DSENet performs the best among three models.

**Glucose Prediction Results.** Table 1 shows the average results for different forecasting lengths in two datasets. It demonstrates that our proposed DSENet achieves stable and near **SOTA** performance on both datasets in multi-time-scale. This indicates that the model is not only suitable for short-term clinical glucose warnings but also capable of reliably supporting long-term trend assessment. Combined with its concise architecture and linear computational complexity, we believe that DSENet has strong potential to serve as a new novel baseline model for blood glucose prediction.

### 5.2. Visualization Results Analysis

To compare the blood glucose prediction results across different models, we visualize the predicted and ground-truth curves of the three best-performing models: Ours, SST, and Crossformer. We consider two forecast horizons: $F = 0.5h$ and $F = 1h$ as examples, and conduct visualization analysis on both datasets. The results are shown in Figure 9. From the visualization results, it can be observed that our model achieves the best forecasting performance compared to other models in the field. Figure 10 and Figure 14 show the visulazation results of DSENet. To ensure experimental consistency, we uniformly fix the historical window to $2 hours$ and predict future glucose levels over multiple lengths.

### 5.3. Ablation Study

We conduct systematic ablation study to verify the contribution of each key component to the overall performance. The advantages of the model in multi-scale forecasting mainly stem from the proposed dual-stream enhancement mech-

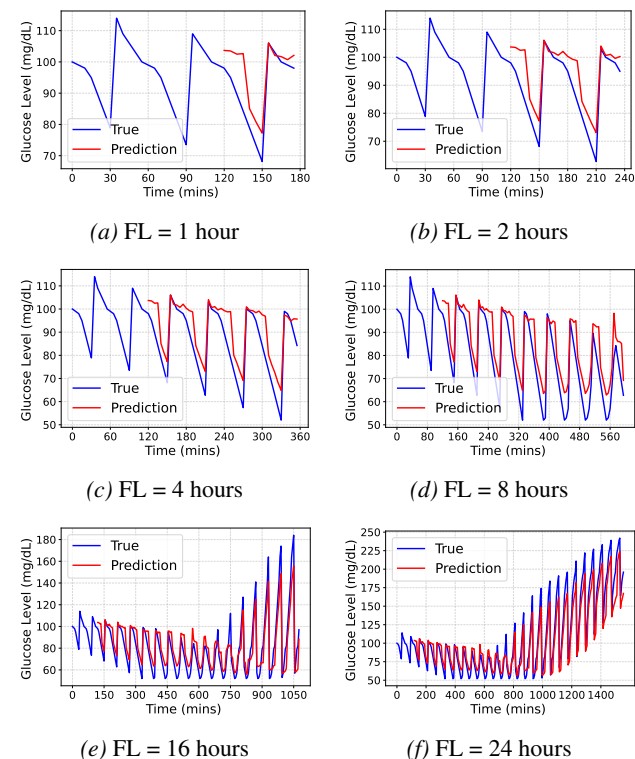

*(a)* FL = 1 hour    *(b)* FL = 2 hours

*(c)* FL = 4 hours    *(d)* FL = 8 hours

*(e)* FL = 16 hours    *(f)* FL = 24 hours

*Figure 10.* Visualization results on OhioT1DM dataset of DSENet.

anism, whose individual effectiveness has been validated in Figure 5 and Figure 7 respectively. The ablation study results are shown in Table 2, removing any component leads to performance degradation, indicating that all components play complementary roles in multi-scale modeling. More ablation study will show in **Appendix** F.

### 5.4. CEG Visualization Analysis

The Clarke Error Grid (CEG) analysis is a clinically established evaluation protocol for assessing the accuracy of predicted blood glucose values against ground-truth measurements and is widely adopted in blood glucose prediction tasks. It compares predictions and references in a two-dimensional space and categorizes prediction errors into five regions. Regions A and B correspond to clinically accurate or acceptable predictions with minimal impact on patient management, whereas Regions C–E indicate poor predictive performance and potential clinical risk, necessitating further model refinement. Taking the 0.5-hour and 1-hour horizons as representative examples, we visualize the corresponding CEG plots for both datasets in Figure 11. It can be observed that nearly all prediction points fall within Region A, with a small fraction located in Region B and none appearing in Regions C–E. Furthermore, we consider three clinically meaningful prediction horizons, $F \in \{0.5\,\text{h}, 1\,\text{h}, 2\,\text{h}\}$, on both datasets. For each setting, we compute the proportion

*Table 2.* Ablation study on two datasets under three different historical lengths and three different forecasting lengths.

| | OhioT1DM | | | DCLP5 | | |
|---|---|---|---|---|---|---|
| | HL = 1h | HL = 2h | HL = 4h | HL = 1h | HL = 2h | HL = 4h |
| **Ours** | **0.389** | **0.394** | **0.390** | **0.455** | **0.460** | **0.476** |
| w/o Bidirection | 0.391 | 0.395 | 0.392 | 0.457 | 0.466 | 0.477 |
| w/o forget gate | 0.399 | 0.417 | 0.421 | 0.470 | 0.468 | 0.500 |
| w/o LoRE | 0.391 | 0.423 | 0.426 | 0.472 | 0.468 | 0.487 |
| w/o DSSAF | 0.397 | 0.403 | 0.400 | 0.470 | 0.475 | 0.488 |
| w/o DMSP | 0.397 | 0.408 | 0.404 | 0.469 | 0.498 | 0.496 |
| | OhioT1DM | | | DCLP5 | | |
| | FL = 0.5h | FL = 1h | FL = 2h | FL = 0.5h | FL = 1h | FL = 2h |
| **Ours** | **0.149** | **0.239** | **0.389** | **0.187** | **0.298** | **0.455** |
| w/o Bidirection | 0.158 | 0.257 | 0.391 | 0.188 | 0.300 | 0.457 |
| w/o forget gate | 0.150 | 0.255 | 0.412 | 0.188 | 0.300 | 0.466 |
| w/o LoRE | 0.152 | 0.246 | 0.393 | 0.196 | 0.316 | 0.472 |
| w/o DSSAF | 0.151 | 0.252 | 0.393 | 0.195 | 0.309 | 0.470 |
| w/o DMSP | 0.150 | 0.250 | 0.408 | 0.196 | 0.319 | 0.469 |

of samples falling into each CEG region and report the percentage of Region A, with results summarized in Figure 12. As shown, our model consistently achieves the highest proportion in Region A across all prediction horizons.

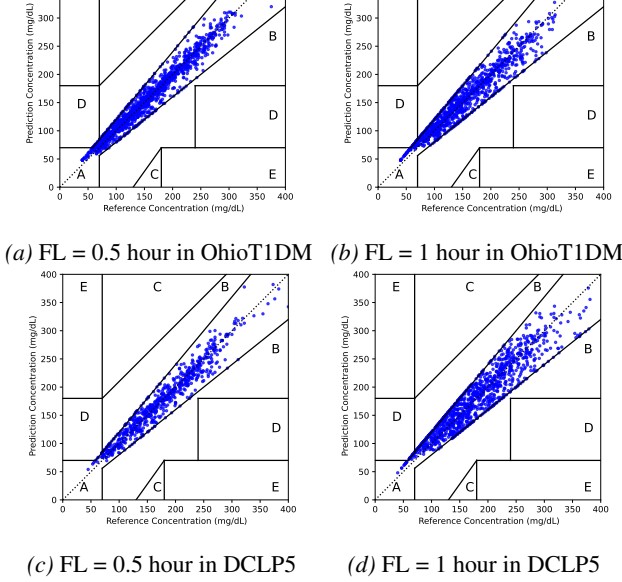

*(a)* FL = 0.5 hour in OhioT1DM   *(b)* FL = 1 hour in OhioT1DM

*(c)* FL = 0.5 hour in DCLP5   *(d)* FL = 1 hour in DCLP5

*Figure 11.* The CEG results of DSENet on OhioT1DM and DCLP5 datasets with two clinically meaningful forecasting lengths.

### 5.5. Model Efficiency

We have theoretically proofed that DSENet maintains linear complexity. Furthermore, we quantitatively evaluate the training latency and GPU memory consumption, with the results illustrated in Figure 13. All models are trained on NVIDIA RTX 4090 GPU. The experimental results demonstrate that DSENet delivers SOTA performance with significant low GPU memory usage and training time. In contrast, our model shows almost no sharp increase in training latency when processing long input series, indicating strong scala-

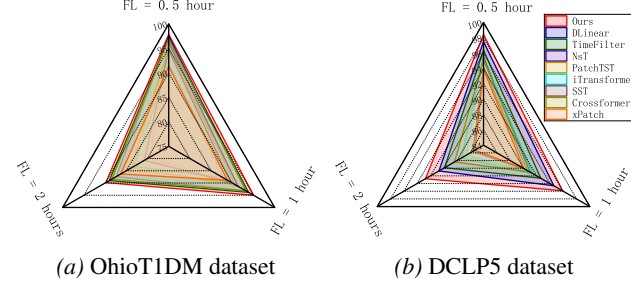

*(a)* OhioT1DM dataset   *(b)* DCLP5 dataset

*Figure 12.* The percentage of Region A in CEG on both datasets. Results show our DSENet performs the best compared with others.

bility and stability. Unlike other methods require reducing model capacity or input length to meet memory constraints, our model can be trained without any performance compromise, highlighting its feasibility and practical deployment potential as a new baseline for blood glucose prediction.

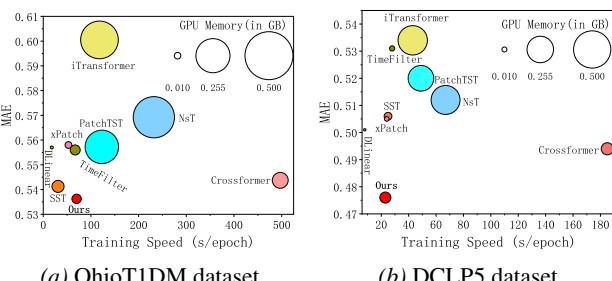

*(a)* OhioT1DM dataset   *(b)* DCLP5 dataset

*Figure 13.* Model efficiency comparison on two datasets.

## 6. Conclusion

In this paper, we propose a novel dual-stream model, DSENet, which explicitly enlarges the representational discrepancy between the global stream and the local stream through the proposed dual-stream enhancement mechanism. This design effectively improves the model's ability to capture local variations. Experimental results demonstrate that DSENet achieves SOTA performance on two public blood glucose prediction datasets and exhibits strong generalization and robustness across multiple real-world time-series datasets. Because of its extremely low computational cost and high sensitivity to local abrupt changes, DSENet shows strong potential to serve as a novel baseline model for blood glucose forecasting. The future work includes exploring its applications in a broader range of biomedical time series tasks and in the field of multi-modal fusion.

## Impact Statement

This paper presents work whose goal is to advance the field of Machine Learning. There are many potential societal consequences of our work, none which we feel must be

specifically highlighted here.

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

## A. Dynamic Temporal Granularity

To quantitatively distinguish the coarse-grained features emphasized by the global pattern from the fine-grained features emphasized by the local pattern, we introduce the concept of dynamic temporal granularity, denoted as $\zeta$. We further use this granularity to guide the selection of patch length and stride for different forecast horizons. What we expect is that when the forecast length is long, the granularity should decrease, indicating that global modeling is performed in a coarser manner. Conversely, when the forecast horizon is short, the granularity should increase, reflecting a finer-grained perception of local variations.

On the other hand, we argue that $\zeta$ should also be related to the length of the original series $L$, patch length $P$ and the number of patches $N$. This is intuitive, since the patching operation decomposes the original sequence into $N$ patches with each patch's length is $P$, and the number of patches determines the effective input received by the model during training. Assuming a fixed forecast length, we consider the following two extreme cases:

(1) **Point-wise.** Each time step of the original time series is individually fed into the subsequent model for processing. In this case, the corresponding value of $\zeta$ should be maximized, as the model focuses on each time step from a highly local perspective and captures fine-grained temporal information. Point-wise patching is equivalent to setting $P = S = 1$ where the number of patches equals the original sequence length, i.e., $N = L$.

(2) **Series-wise.** The entire original sequence is fed into the subsequent model as a whole for global processing. In this case, the corresponding value of $\zeta$ should be minimized, since the model operates from a macro-level perspective without attending to individual time steps. Series-wise patching can be viewed as performing no patching operation, where the number of patches is $N = 1$.

Based on the analysis of these t wo extreme cases and the inherent monotonicity of $\zeta$, we define $\zeta$ as follows:

**Definition A.1.** $\zeta$ can be generated by the following expression that related to series length $L$, patch numbers $N$ and forecast length $F$: $\zeta_i(F) = \Phi\big(L, \ P_i(F), \ S_i(F)\big) = \frac{N_i}{L_i \cdot P_i^\gamma}, \quad i \in \{g, l\}$

The power-law form in Definition A.1 satisfies the analysis discussed above.

**Lemma A.2.** $\gamma = \frac{1}{2}$

*Proof.* We use Fisher information to characterize the effective information content of each token. We assume that the signal within a patch can be approximated by a locally constant parameter $x_t = \theta + \varepsilon_t$ with distribution $\varepsilon_t \overset{\text{i.i.d.}}{\sim} \mathcal{N}(0, \sigma^2)$, where i.i.d refers to the independent and identically distribution:

$$p(\mathbf{x} \mid \theta) = \prod_{t=1}^{P} \frac{1}{\sqrt{2\pi\sigma^2}} \exp\left(-\frac{(x_t - \theta)^2}{2\sigma^2}\right) \tag{4}$$

Take the logarithm yields the log-likelihood as follow:

$$\ell(\theta) = \log p(\mathbf{x} \mid \theta) = -\frac{P}{2} \log(2\pi\sigma^2) - \frac{1}{2\sigma^2} \sum_{t=1}^{P} (x_t - \theta)^2 \tag{5}$$

By differentiating the likelihood function in Equation (5), we obtain the score function:

$$\frac{\partial \ell(\theta)}{\partial \theta} = -\frac{1}{2\sigma^2} \sum_{t=1}^{P} \frac{\partial}{\partial \theta} (x_t - \theta)^2 = -\frac{1}{2\sigma^2} \sum_{t=1}^{P} 2(\theta - x_t) = \frac{1}{\sigma^2} \sum_{t=1}^{P} (x_t - \theta) \tag{6}$$

where $\frac{\partial \ell(\theta)}{\partial \theta}$ represents the score.

For Fisher information with respect to the scalar parameter $\theta$, there are two commonly used definitions: one based on the variance of the score function, and the other based on the expectation of the negative second derivative of the log-likelihood. In this work, we adopt the latter definition for computation, noting that the two definitions are theoretically equivalent. The expectation-based definition of the negative second derivative is given by

$$\mathcal{I}(\theta) = -\mathbb{E}\left[\frac{\partial^2 \ell(\theta)}{\partial \theta^2}\right] \tag{7}$$

*Table 3.* Some Hyper-parameters used in Dynamic Multi-Scale Patching (DMSP).

| Hyper-parameter | Value |
|---|---|
| $\beta_{\min}$ | 0.5 |
| $\beta_{\max}$ | 6 |
| $F^*$ | 12 |
| $P$ | $P_{min} = 2, P_{max} = L$ |
| $S$ | $S_{min} = 1, S_{max} = P$ |
| For $F \leq F^*$ | $P_g^{(0)} = 2, S_g^{(0)} = 2, d_{\mathrm{conv}} = 2$ |
| For $F \geq F^*$ | $P_l^{(0)} = 2 \; S_l^{(0)} = 1, d_{\mathrm{conv}} = 6$ |
| $\alpha_p, \alpha_s$ | $\alpha_p \in [2, 32], \alpha_s \in [1, 16]$ **Note:** Larger $L$, Larger $\alpha_p$ and $\alpha_s$ |

Taking the second derivative of Equation (7), we can get:

$$\frac{\partial^2 \ell(\theta)}{\partial \theta^2} = \frac{\partial}{\partial \theta}\left(\frac{1}{\sigma^2}\sum_{t=1}^{P}(x_t - \theta)\right) = \frac{1}{\sigma^2}\sum_{t=1}^{P}(-1) = -\frac{P}{\sigma^2} \tag{8}$$

Take the formula in Equation (8) back to Equation (7), the Fisher information will be defined as:

$$\mathcal{I}_P(\theta) = -\mathbb{E}\left[\frac{\partial^2 \ell(\theta)}{\partial \theta^2}\right] = -\left(-\frac{P}{\sigma^2}\right) = \frac{P}{\sigma^2} \tag{9}$$

As a result, we can conclude that the Fisher information of a patch with length $P$ with respect to parameter $\theta$ is $\frac{P}{\sigma^2}$, which grows linearly with $P$.

Since the Fisher information itself is $\mathcal{O}(P)$, we further introduce the Cramér–Rao Lower Bound (CRLB), which states that the variance lower bound of any unbiased estimator of $\hat{\theta}$ is:

$$\mathrm{Var}(\hat{\theta}) \geq \frac{1}{\mathcal{I}_P(\theta)} = \frac{\sigma^2}{P} \tag{10}$$

Taking the standard deviation form yields:

$$\mathrm{Std}(\hat{\theta}) \geq \frac{\sigma}{\sqrt{P}} \tag{11}$$

which is commonly interpreted as the natural scale of distinguishable precision. Therefore, choosing $\gamma = \frac{1}{2}$ is reasonable, as it allows the granularity $\zeta$ to incorporate a power-law dependence on $P$.

Now we use Lemma A.2 to re-write the expression in Theorem A.1, that is:

$$\zeta_i(t) = \frac{N_i}{L_i P_i^{\frac{1}{2}}} = \left(\left\lfloor \frac{L - P_i}{S_i} \right\rfloor + 1\right) \Big/ \left(L_i P_i^{\frac{1}{2}}\right) \approx \frac{1}{S_i \sqrt{P_i}} \tag{12}$$

Then we bring Equation (1) to Equation (12), we can get:

$$\zeta_i(F) = \frac{1}{S_i \sqrt{P_i}} = \frac{1}{\alpha_s \sqrt{\alpha_p} S^{(0)} \sqrt{P^{(0)}}} \cdot \beta(F)^{-\frac{3}{2}} = c \cdot \beta(F)^{-\frac{3}{2}} \tag{13}$$

where $c$ is the constant number related to $\alpha_s$, $\alpha_p$, $S^{(0)}$ and $P^{(0)}$.

Therefore, we successfully demonstrate that Definition 4.2 is well justified. The resulting formulation of dynamic temporal granularity $\zeta$ has strong practical significance. Specifically, it depends solely on the forecast length and is monotonically decreasing with respect to $F$, which is consistent with the conclusion presented in the main text: longer forecast horizons should correspond to smaller granularity, representing coarser global features, while shorter forecast horizons should correspond to larger granularity, representing finer local fluctuations.

The hyper-parameters in this paper are shown in Table 3. When the forecast length is fixed, the granularity can be determined through empirically motivated computation in Table 3. Thus, a range is limited for the patch length and step size under a specific predicted length, and guidance is provided for specific selection. In this way, a reasonable pair of $(P_i, S_i)$ can be assigned to each forecast length.

## B. Why the *LoRE* Block Enhances Local Perception?

The LoRE block is applied before the self-attention module with the purpose of further enhancing the local stream's sensitivity to local abrupt changes, thereby achieving the goal of dual-stream enhancement. Recall that the input tensor after patching has the shape $x \in \mathbb{R}^{B \times M \times N \times P}$, where $B$ is batch size. We denote the corresponding patch vector as $u \in \mathbb{R}^P$. The LoRE block applies a shared feed-forward network and a dynamic 1D convolution on the last dimension.

### B.1. FFN as a Low-Rank Enhancement of the Identity

The whole process of FFN can be summarized as:

$$\text{FFN}(u) = W_2 \, \phi(W_1 u) \tag{14}$$

where $W_1 \in \mathbb{R}^{r \times P}$ and $W_2 \in \mathbb{R}^{P \times r}$, $r$ is the inner dimension of model, $\phi$ is a pointwise nonlinearity function. There is also a residual connection of FFN, thus the final process comes to : $f(u) = u + FFN(u)$.

**Lemma B.1.** *The Jacobian Matrix of $FFN(u)$ has the ceiling rank of $r$, i.e.* $\text{rank}\big(J_{\text{FFN}}(u)\big) \leq r$

*Proof.* The FFN is a composition of a linear map, a pointwise nonlinearity and another linear map. For the Jacobian Matrix of $FFN(u)$, by using chain rule, we can get:

$$J_{\text{FFN}}(u) = W_2 \, \text{Diag}\big(\phi'(W_1 u)\big) \, W_1 \tag{15}$$

As a result, the product of a $P \times r$ matrix, a diagonal $r \times r$ matrix and a $r \times P$ matrix will result in a Jacobian Matrix of $FFN$ with rank at most $r$.

For the final process, the Jacobian Matrix of $f(u)$ is $J_f(u) = I_P + J_{\text{FFN}}(u)$, where $I_P$ is the $P \times P$ identity matrix of $u$.

**Theorem B.2.** *For any $u \in \mathbb{R}^L$, it satisfies $J_f(u) - I_L = J_{\text{FFN}}(u)$ and $\text{rank}\big(J_f(u) - I_L\big) \leq r$*

It is obvious. In particular, when $r < P$, the deviation of $f$ from the identity mapping is a $rank - r$ perturbation. In the sense of Jacobian, each patch is transformed by the identity matrix plus a low-rank operator in the patch dimension. In this way, The $FFN$ induces input deviations along the patch dimension, corresponding to a low-rank enhancement with rank no greater than $r$. What's more, layer normalization is added after the $FFN$. Layer normalization is an affine, per-sample re-scaling on $\mathbb{R}^L$, its Jacobian at a given point is a full matrix with bounded operator norm. As it is applied after the residual addition, it does not change the fact that the non-identity component of the mapping on the patch dimension is $rank - r$.

### B.2. Dynamic Convolution as a Low-dimensional Operator Family

We now consider another important submodule: Dynamic Convolution. The input is $x \in \mathbb{R}^{N \times P}$, for each patch $p$, we write the 1D signal as $x_p \in \mathbb{R}^P$. The dynamic convolution constructs an input-independent kernel as a mixture of $K$ base kernels: $\{w_k\}_{k=1}^K \subset \mathbb{R}^D$, where $D = 1 \times kernelsize$. Let $W \in \mathbb{R}^{K \times D}$ be the matrix whose rows are the flattened base kernels $w_k^\top$. For each patch, we define the weight vector $\alpha(x_p) \in \Delta^{K-1}$ (the probability simplex) via:

$$\alpha(x_p) = \text{softmax}(A_2 \, \phi(A_1 x_p)/\tau) \tag{16}$$

and the effective kernel is:

$$w(x_p) = \sum_{k=1}^K \alpha_k(x_p) \, w_k = W^\top \alpha(x_p) \in \mathbb{R}^D \tag{17}$$

**Lemma B.3.** *Let $\mathcal{H} \subset \mathbb{R}^D$ be the set of all kernels that can realized by Dynamic Convolution: $\mathcal{H} = \{\, w(x_p) : x_p \in \mathbb{R}^L \,\}$. Then it has $\mathcal{H} \subset \text{span}\{w_1, \ldots, w_K\}$ and $\dim \mathcal{H} \leq \text{rank}(W) \leq \min\{K, D\}$*

*Proof.* By construction, every kernel $w(x_p)$ is a linear combination of the base kernels with coefficients $\alpha(x_p)$, as shown in Equation (17). Thus, $\mathcal{H} \subset \text{span}\{w_1, \ldots, w_K\}$. The dimension of this span is the rank of $W$, which is at most $\min\{K, D\}$. Furthermore, we represent the convolution operator as a Toeplitz Matrix, which allows us to analysis its spectral and rank properties using linear algebra tools. The matrix $T(w) \in \mathbb{R}^{P \times P}$ is generated from kernel $w$. The image of dynamic convolution in the space of linear operators $\mathbb{R}^P \to \mathbb{R}^P$ is as follow:

$$\mathcal{T} = \{\, T(w(x_p)) : x_p \in \mathbb{R}^P \,\} \tag{18}$$

**Theorem B.4.** *Let $\mathcal{T}_0 = \mathrm{span}\{T(w_1), \ldots, T(w_K)\}$. Then $\mathcal{T} \subseteq \mathcal{T}_0$ and $\dim \mathcal{T}_0 \leq \min\{K, D\}$*

*Proof.* For any input $x_p$, the convolution operator is $T\big(w(x_p)\big)$ with $w(x_p) \in \mathrm{span}\{w_k\}$. By linearity of the Toeplitz construction, $T\big(w(x_p)\big) \in \mathrm{span}\{T(w_k)\} = \mathcal{T}_0$. The dimension bound follows from Lemma B.3.

Hence, the dynamic convolution does not search over the entire space of convolution operators, but over a low-dimensional subspace spanned by $K$ basis operators. When $K \ll D$, this provides a strong low-rank inductive bias in the space of local filters, while still allowing input-dependent adaptation through $\alpha(x_p)$. In other words, the presence of dynamic convolution restricts the model to focus only on the subspace formed by a few neighboring tokens, which substantially enhances the local perception capability of the LoRE block.

In conclusion, we combine the FNN part and the dynamic convolution part to get: $\mathrm{LoRE}(u) = T\big(w(u)\big) f(u)$. We then perform linearization in the neighborhood of $\bar{u}$, whose first-order Taylor expansion is given by:

$$\mathrm{LoRE}(u) \approx \mathrm{LoRE}(\bar{u}) + J_{\mathrm{LoRE}}(\bar{u})(u - \bar{u}) \tag{19}$$

Recall that the definition of Jacobian is $J_{\mathrm{LoRE}}(\bar{u}) = \left.\frac{\partial \mathrm{LoRE}(u)}{\partial u}\right|_{u=\bar{u}}$. Therefore, it suffices to compute the derivative of $\mathrm{LoRE}(u)$. By applying the product rule, we obtain:

$$J_{\mathrm{LoRE}}(u) = T\big(w(u)\big) J_f(u) + \underbrace{\frac{\partial T(w(u))}{\partial u} f(u)}_{\text{kernel-adaptation term}} \tag{20}$$

For the second term in Equation (20), it can denote the following formula by using chain rule:

$$\frac{\partial T(w(u))}{\partial u} f(u) = \left(\frac{\partial T}{\partial w}\right) \nabla w(u) \, f(u) \tag{21}$$

As a result, it satisfies the Theorem 4.3 mentioned above. Intuitively, the FFN introduces low-rank features into the sequence, while dynamic convolution encourages the model to focus more on local variations. The combination of these two components effectively enhances the model's local perception capability.

Specifically, based on all the theoretical analyses presented above, we can summarize the following: The Jacobian of LoRE reveals its enhanced local sensitivity: while the leading term corresponds to a local convolutional operator, the additional kernel-adaptation term introduces input-dependent amplification for localized perturbations. As a result, small but abrupt variations within a patch induce disproportionately larger responses, which in turn shifts the spectral energy toward higher frequencies and reduces the low-frequency leakage.

## C. Effectiveness Analysis of the Dual-Stream Enhancement Mechanism

In this paper, we propose a novel mechanism termed the ***Dual-Stream Enhancement Mechanism***. Its core objective is to explicitly enlarge the discrepancy between the paradigms emphasized by the two streams, enabling each stream to focus on its designated role while reducing unnecessary interaction between them. Specifically, we introduce structural improvements to the dual-stream architecture. In the global stream, we employ the DFB-Mamba block with stronger memory capacity to model global features of the sequence. In the local stream, we first apply the LoRE block to amplify local fluctuations, and then use a self-attention mechanism to model local features, thereby enhancing local perception capability.

To theoretically justify that the proposed dual-stream enhancement mechanism effectively enlarges the discrepancy between the two paradigms, we conduct analysis from the perspectives of signal processing and functional analysis.

Recall that the input series of the two streams is $x \in \mathbb{R}^{M \times L}$, after DMSP Alg., it can get the patching series: $X_g \in \mathbb{R}^{B \times M \times N_g \times P_g}$ for global stream and $X_l \in \mathbb{R}^{B \times M \times N_l \times P_l}$. For the global stream, we use DFB-Mamba block, the final output is $g = G_\theta(X_g) \in \mathbb{R}^{B \times M \times N_g \times D}$, where $D$ is the dimension of Mamba. For the local stream, we use LoRE followed by Light-Weight self-attention encoder, the final output is $\ell = L_\phi(X_\ell) \in \mathbb{R}^{B \times M \times N_\ell \times P_l}$. For the sake of analysis, we adopt a standard local linearization assumption:

**Assumption C.1.** In a neighborhood of the data manifold $\mathcal{M}$, the global stream and the local stream of DSENet admit a first-order local linearization. Specifically, for a small perturbation $\delta x$, we have $G_\theta(x + \delta x) - G_\theta(x) \approx \mathcal{G}_x \, \delta x$ for global

stream and $L_\phi(x + \delta x) - L_\phi(x) \approx \mathcal{L}_x \, \delta x$ for local stream, where $\mathcal{G}_x$ and $\mathcal{L}_x$ are input-dependent linear operators(Jacobians). To enable a frequency-domain interpretation, we further approximate $\mathcal{G}_x$ and $\mathcal{L}_x$ as quasi-static and locally shift-invariant over short temporal neighborhoods, and model them by effective convolution operators: $(\mathcal{G}_x \delta x)_t \approx \sum_\tau K_g(\tau; x) \, \delta x_{t-\tau}$ for global stream and $(\mathcal{L}_x \delta x)_t \approx \sum_\tau K_\ell(\tau; x) \, \delta x_{t-\tau}$ for local stream, where $K_g(\cdot; x)$ and $K_\ell(\cdot; x)$ denote the input-condiationed effective kernel.

### C.1. Global Stream (Can be Viewed as a Low-Pass Filter)

We use Mamba-based model in the global stream. The core of Mamba is a discrete-time state-space model. When a unit impulse is provided as input, the corresponding impulse response of the system is given by:

$$k_{\text{ssm}}(t) = \begin{cases} CA^{t-0}B + D\delta_{t0}, & t \geq 0, \\ 0, & t < 0. \end{cases} \tag{22}$$

In the Mamba model, the standard condition for SSM stability is that the spectral radius of matrix $A$ is less than 1 ($\rho(A) < 1$), which is also referred to as BIBO stable. From linear algebra, we know that there exists a constant $C_0 > 0, \quad \rho \in (0,1)$ let $\|A^t\| \leq C_0\rho^t, \quad t \geq 0$. Thus, it satisfies $|k_{\text{ssm}}(t)| \leq \|C\| \, \|A^t\| \, \|B\| + |D|\delta_{t0} \leq C_0\rho^t$. It indicates that the impulse response decays exponentially with respect to $t$, which guarantees that: $\sum_t |k_{\text{ssm}}(t)| < \infty, \ \sum_t |k_{\text{ssm}}(t)|^2 < \infty$. That is to say, the response $k$ is in $\ell^1 \cap \ell^2$ space.

*Note:* Vanilla Mamba has such properties, for our proposed DFB-Mamba, it also satisfies the properties above. Because they merely apply bounded coefficients to weight the output at each time step.

According to the discrete time Fourier transform, the frequency response is:

$$H_g(\omega) = \sum_{\tau=-\infty}^{\infty} K_g(\tau) \, e^{-j\omega\tau} \tag{23}$$

Since $K_g(\tau) \in \ell^1$, the series converges uniformly. Therefore, $H_g(\omega)$ is a continuous function with finite magnitude. Furthermore, since $K_g(\tau) \in \ell^2$, the discrete-time Parseval identity holds:

$$\sum_{\tau=-\infty}^{\infty} |K_g(\tau)|^2 = \frac{1}{2\pi} \int_{-\pi}^{\pi} |H_g(\omega)|^2 \, d\omega. \tag{24}$$

The right-term is finite, thus:

$$E_{\text{tot}} := \int_{-\pi}^{\pi} |H_g(\omega)|^2 \, d\omega < \infty. \tag{25}$$

In conclusion, the spectral energy is finite. So we can view this problem from the perspective of energy under this premise and give the following definition.

**Definition C.2.** We define the high-frequency energy: $E_{\text{out}}(\omega_c) := \int_{|\omega|>\omega_c} |H_g(\omega)|^2 \, d\omega$ which means the energy of the spectral components whose absolute frequencies are greater than $\omega_c$. Furthermore, we define another metric: $\varepsilon_g(\omega_c) := \frac{E_{\text{out}}(\omega_c)}{E_{\text{tot}}} = \frac{\int_{|\omega|>\omega_c} |H_g(\omega)|^2 \, d\omega}{\int_{-\pi}^{\pi} |H_g(\omega)|^2 \, d\omega}$, which means the proportion of the total energy contained in frequencies higher than $\omega_c$.

Based on the Definition C.2, the $\varepsilon_g(\omega_c)$ obeys the following Theorem.

**Theorem C.3.** (1) $0 \leq \varepsilon_g(\omega_c) \leq 1$. (2) $\varepsilon_g(\omega_c)$ *is monotonically non-increasing of* $\omega_c$. (3) $\lim_{\omega_c \to \pi} \varepsilon_g(\omega_c) = 0$.

*Proof.* (1) is obvious. For (2), if $\omega_{c,1} < \omega_{c,2}$, then it has: $\{ |\omega| > \omega_{c,2} \} \subset \{ |\omega| > \omega_{c,1} \}$. That is, as the cut-off frequency increases, more of the intermediate low-frequency components are removed, while the remaining high-frequency tail becomes increasingly smaller; therefore:

$$E_{\text{out}}(\omega_{c,2}) = \int_{|\omega|>\omega_{c,2}} |H_g(\omega)|^2 \, d\omega \leq \int_{|\omega|>\omega_{c,1}} |H_g(\omega)|^2 \, d\omega = E_{\text{out}}(\omega_{c,1}). \tag{26}$$

Divide the constant $E_{tot}$ simultaneously on the both sides, it can get $\varepsilon_g(\omega_{c,2}) \leq \varepsilon_g(\omega_{c,1})$.

For (3), when $\omega_c \to \pi$, the interval $\{ |\omega| > \omega_c \}$ becomes increasingly narrow and eventually shrinks to the empty set. Also, $|H_g(\omega)|^2$ is an integrable function.

**Lemma C.4.** *If $f \in L^1([-\pi, \pi])$, for any family of intervals $A_\alpha$ that monotonically decreases and shrinks to the empty set, it has $\int_{A_\alpha} |f| \to 0$.*

Lemma C.4 is a classical result in classic measure theory and real variable function analysis, the proof is omitted for brevity. Here we let $f(\omega) = |H_g(\omega)|^2$, $A_{\omega_c} := \{\, |\omega| > \omega_c \,\}$. Then $\omega_c \to \pi \Rightarrow A_{\omega_c} \downarrow \varnothing$, thus:

$$\lim_{\omega_c \to \pi} E_{\text{out}}(\omega_c) = \lim_{\omega_c \to \pi} \int_{|\omega| > \omega_c} |H_g(\omega)|^2 \, d\omega = 0. \tag{27}$$

Therefore, we successfully demonstrate that the DFB-Mamba model in the global stream is essentially a low-pass–filter-like architecture. As the frequency increases, the energy carried by its frequency band decreases significantly, indicating that DFB-Mamba is more suitable for capturing smooth global trends. When modeling global features, the energy contained in its frequency band is correspondingly larger.

### C.2. Local Stream (Can be Viewed as a High-Pass Filter)

In contrast to the global stream, the local stream is expected to focus on the high-frequency components of the sequence, as illustrated in Figure 2. Therefore, combined with the previous energy analysis of the global stream, we can directly define the local stream as follows:

**Definition C.5.** Similar to global stream, we give a direct definition of $\varepsilon_\ell(\omega_c)$, $\varepsilon_\ell(\omega_c) = \dfrac{\displaystyle\int_{|\omega| < \omega_c} |H_\ell(\omega)|^2 \, d\omega}{\displaystyle\int_{-\pi}^{\pi} |H_\ell(\omega)|^2 \, d\omega}$ It has the

monotonically non-decreasing property because it can be viewed as a high-pass-filter which is at the opposite side of global stream.

### C.3. Difference Between Global and Local Stream

To quantitatively characterize the discrepancy between the global stream and the local stream, we introduce the following definition.

**Definition C.6.** We define a new parameter $\mathcal{O}_{\text{spec}}$ to measure the spectral overlap between the global stream and the local

stream, the expression is: $\mathcal{O}_{\text{spec}}(G_\theta, L_\phi) = \dfrac{\displaystyle\int_{-\pi}^{\pi} |H_g(\omega)| \, |H_\ell(\omega)| \, d\omega}{\left(\displaystyle\int_{-\pi}^{\pi} |H_g(\omega)|^2 \, d\omega\right)^{1/2} \left(\displaystyle\int_{-\pi}^{\pi} |H_\ell(\omega)|^2 \, d\omega\right)^{1/2}}$. Then the difference between

the two is denoted as: $D_{\text{spec}}(G_\theta, L_\phi) = 1 - \mathcal{O}_{\text{spec}}(G_\theta, L_\phi)$

Let $E_g := \int_{-\pi}^{\pi} |H_g(\omega)|^2 \, d\omega$ and $E_\ell := \int_{-\pi}^{\pi} |H_\ell(\omega)|^2 \, d\omega$ for convenient. We have already demonstrated that global stream is more like a low-pass filter and local stream is more like a high-pass filter. Thus, for low-pass filter, it has $\int_{|\omega| > \omega_c} |H_g(\omega)|^2 \, d\omega \leq \varepsilon_g \, E_g$ and for high-pass filter, it has $\int_{|\omega| < \omega_c} |H_\ell(\omega)|^2 \, d\omega \leq \varepsilon_\ell \, E_\ell$.

**Theorem C.7.** $O_{\text{spec}}(G_\theta, L_\phi) \leq \sqrt{\varepsilon_g} + \sqrt{\varepsilon_\ell}$

*Proof.* We consider explicitly partitioning the integration domain into a low-frequency domain and a high-frequency domain. We denote them as $A := \{\, \omega \in [-\pi, \pi] : |\omega| \leq \omega_c \,\}$ and $B := \{\, \omega \in [-\pi, \pi] : |\omega| > \omega_c \,\}$ separately. We decompose the numerator of $\mathcal{O}_{\text{spec}}$ into two components: $I := \int_{-\pi}^{\pi} |H_g(\omega)| \, |H_\ell(\omega)| \, d\omega = I_A + I_B$, where $I_A := \int_A |H_g(\omega)| \, |H_\ell(\omega)| \, d\omega$ and $I_B := \int_B |H_g(\omega)| \, |H_\ell(\omega)| \, d\omega$. We first consider the low-frequency domain.

In the domain $A$, we apply the Cauchy-Schwarz Inequality:

$$I_A = \int_A |H_g(\omega)| \, |H_\ell(\omega)| \, d\omega \leq \left(\int_A |H_g(\omega)|^2 \, d\omega\right)^{1/2} \left(\int_A |H_\ell(\omega)|^2 \, d\omega\right)^{1/2} \tag{28}$$

For $H_g$, it has:

$$\int_A |H_g(\omega)|^2 \, d\omega \leq \int_{-\pi}^{\pi} |H_g(\omega)|^2 \, d\omega = E_g \tag{29}$$

For $H_l$, we notice that $A = \{\, |\omega| \le \omega_c \,\} \subset \{\, |\omega| < \omega_c \,\} \cup \{\, \omega = \pm\omega_c \,\}$ Since a singleton set has zero measure under the **Lebesgue** integral, it does not contribute to the integral, and thus the high-pass property can be directly applied:

$$\int_A |H_\ell(\omega)|^2 \, d\omega \le \int_{|\omega| < \omega_c} |H_\ell(\omega)|^2 \, d\omega \le \varepsilon_\ell \, E_\ell. \tag{30}$$

Then we bring Equation (30) and Equation (29) back to Equation (28), we can get the final expression:

$$I_A \le \sqrt{E_g} \, \sqrt{\varepsilon_\ell E_\ell} = \sqrt{\varepsilon_\ell} \, \sqrt{E_g E_\ell}. \tag{31}$$

In the domain $B$, we do the same operation as above. It can get:

$$I_B \le \sqrt{\varepsilon_g E_g} \, \sqrt{E_\ell} = \sqrt{\varepsilon_g} \, \sqrt{E_g E_\ell}. \tag{32}$$

Then we combine Equation (31) and Equation (32) to get:

$$I = I_A + I_B \le \sqrt{\varepsilon_\ell} \, \sqrt{E_g E_\ell} + \sqrt{\varepsilon_g} \, \sqrt{E_g E_\ell} = \left(\sqrt{\varepsilon_\ell} + \sqrt{\varepsilon_g}\right) \sqrt{E_g E_\ell}. \tag{33}$$

Then we bring Equation (33) back to the definition expression of $\mathcal{O}_{\mathrm{spec}}$, it turns to:

$$\mathcal{O}_{\mathrm{spec}}(G_\theta, L_\phi) = \frac{I}{\sqrt{E_g} \, \sqrt{E_\ell}} \le \frac{\left(\sqrt{\varepsilon_\ell} + \sqrt{\varepsilon_g}\right) \sqrt{E_g E_\ell}}{\sqrt{E_g} \, \sqrt{E_\ell}} = \sqrt{\varepsilon_\ell} + \sqrt{\varepsilon_g}. \tag{34}$$

Thus, for $D_{\mathrm{spec}}$, it also has:

$$D_{\mathrm{spec}} \ge 1 - \sqrt{\varepsilon_\ell} - \sqrt{\varepsilon_g} \tag{35}$$

Therefore, we use $D_{\mathrm{spec}}$ to quantitatively characterize the discrepancy between the two paradigms. Although it is formulated as an inequality rather than an equality that directly measures a specific parameter, such a formulation still provides a valid criterion for assessing the discrepancy between the two streams. Specifically, Equation (35) defines a lower bound, representing a theoretically guaranteed minimum. In practice, this bound corresponds to the worst-case scenario. Consequently, to demonstrate that the proposed model substantially enlarges the discrepancy between the two streams, it suffices to show that the right-hand side of the inequality can be increased. Notably, the terms $\varepsilon_\ell$ and $\varepsilon_g$ on the right-hand side are related to the energies of the global and local streams defined in Section C.1 and Section C.2. Leveraging the fact both quantities are monotonic functions, we further derive the substantive proof in Section C.4 under the DSENet structure.

### C.4. Why *DFB-Mamba* and *LoRE* can Enlarge the Difference between Global and Local Stream?

Recalling the analysis in Section C.1, we have shown that the global stream can be interpreted as a low-pass filter, while the local stream behaves as a high-pass filter, and we further introduced a new parameter $D_{\mathrm{spec}}$ to quantitatively measure the discrepancy between the two streams. Although the main paper has already demonstrated, through visualizations, that DFB-Mamba substantially enhances the capability of capturing global characteristics compared to Vanilla Mamba, and that the incorporation of the LoRE block significantly improves the local stream's sensitivity to local variations—further supported by a Jacobian-based analysis in Section B—there is still a lack of a comprehensive theoretical proof. In particular, it remains to be rigorously shown, from a mathematical perspective, that the introduction of DFB-Mamba and the LoRE block directly leads to a significant amplification of the discrepancy between the two streams.

Building upon the preceding analysis, we next provide a further theoretical justification for the effectiveness and soundness of the proposed dual-stream enhancement mechanism.

***For global stream,*** let the original frequency response be denoted by $H(\omega)$. By multiplying it with a frequency weighting function $Q(\omega)$, we obtain a new frequency response:

$$\tilde{H}(\omega) = Q(\omega) \, H(\omega). \tag{36}$$

With respect to the new frequency response, the corresponding high-frequency leakage is:

$$\tilde{\varepsilon}(\omega_c) = \frac{\displaystyle\int_{|\omega| > \omega_c} |Q(\omega)|^2 \, |H(\omega)|^2}{\displaystyle\int_{-\pi}^{\pi} |Q(\omega)|^2 \, |H(\omega)|^2}. \tag{37}$$

*Table 4.* Multi-Scale time series forecasting results across the OhioT1DM dataset. The best is in **bold** and second-best is underlined. FL refers to forecasting length, HL refers to historical length.

| | Model | DSENet (Ours) | | TimeFilter (ICML, 2025) | | xPatch (AAAI, 2025) | | SST (CIKM, 2025) | | DLinear (AAAI, 2023) | | iTransformer (ICLR, 2024) | | Crossformer (ICLR, 2023) | | PatchTST (ICLR, 2023) | | NsT (NeurIPS, 2022) | |
|---|---|---|---|---|---|---|---|---|---|---|---|---|---|---|---|---|---|---|---|
| | Metrics | MSE | MAE | MSE | MAE | MSE | MAE | MSE | MAE | MSE | MAE | MSE | MAE | MSE | MAE | MSE | MAE | MSE | MAE |
| HL=1 hour | FL = 0.5 hour | **0.060** | 0.149 | 0.066 | 0.146 | 0.066 | **0.145** | 0.063 | 0.152 | 0.069 | 0.157 | 0.064 | **0.145** | 0.061 | 0.154 | 0.066 | 0.151 | 0.064 | 0.150 |
| | FL = 1 hour | **0.144** | **0.239** | 0.173 | 0.252 | 0.175 | 0.250 | 0.155 | 0.250 | 0.168 | 0.259 | 0.173 | 0.250 | 0.151 | 0.255 | 0.172 | 0.253 | 0.154 | 0.240 |
| | FL = 2 hours | 0.326 | 0.389 | 0.410 | 0.413 | 0.411 | 0.410 | 0.333 | 0.397 | 0.354 | 0.420 | 0.410 | 0.415 | **0.317** | 0.385 | 0.399 | 0.409 | 0.332 | **0.380** |
| | FL = 4 hours | **0.528** | **0.525** | 0.758 | 0.599 | 0.749 | 0.594 | 0.537 | 0.541 | 0.543 | 0.553 | 0.766 | 0.604 | 0.540 | 0.547 | 0.743 | 0.597 | 0.574 | 0.539 |
| | FL = 8 hours | **0.677** | 0.639 | 1.074 | 0.748 | 1.068 | 0.746 | 0.687 | 0.641 | 0.684 | 0.643 | 1.082 | 0.753 | 0.714 | 0.648 | 1.068 | 0.746 | 0.742 | 0.752 |
| | FL = 16 hours | 0.744 | 0.689 | 1.232 | 0.840 | 1.224 | 0.836 | 0.752 | 0.690 | **0.743** | 0.691 | 1.253 | 0.846 | 0.753 | 0.693 | 1.224 | 0.837 | 0.894 | 0.733 |
| | FL = 24 hours | **0.778** | 0.710 | 1.324 | 0.887 | 1.311 | 0.882 | 0.786 | 0.712 | 0.784 | 0.711 | 1.323 | 0.887 | 0.786 | 0.717 | 1.318 | 0.885 | 1.015 | 0.794 |
| HL=2 hours | FL = 0.5 hour | **0.061** | 0.148 | 0.068 | 0.150 | 0.066 | 0.147 | 0.065 | 0.153 | 0.067 | 0.153 | 0.068 | 0.160 | **0.061** | 0.155 | 0.067 | 0.154 | 0.063 | **0.144** |
| | FL = 1 hour | **0.155** | **0.250** | 0.180 | 0.255 | 0.175 | **0.250** | 0.159 | 0.254 | 0.165 | 0.256 | 0.177 | 0.260 | 0.166 | 0.284 | 0.171 | 0.260 | 0.165 | 0.261 |
| | FL = 2 hours | **0.328** | **0.394** | 0.409 | 0.414 | 0.421 | 0.413 | 0.339 | 0.400 | 0.342 | 0.399 | 0.416 | 0.416 | 0.333 | 0.403 | 0.391 | 0.409 | 0.329 | **0.377** |
| | FL = 4 hours | **0.529** | 0.538 | 0.723 | 0.590 | 0.724 | 0.591 | 0.545 | 0.545 | 0.541 | 0.551 | 0.754 | 0.604 | 0.545 | 0.554 | 0.715 | 0.589 | 0.548 | **0.526** |
| | FL = 8 hours | 0.677 | **0.639** | 1.010 | 0.732 | 1.022 | 0.734 | 0.685 | 0.640 | **0.672** | 0.640 | 1.056 | 0.743 | 0.689 | 0.644 | 1.007 | 0.730 | 0.731 | 0.640 |
| | FL = 16 hours | **0.746** | **0.689** | 1.144 | 0.814 | 1.139 | 0.812 | 0.749 | 0.690 | 0.753 | 0.690 | 1.165 | 0.820 | 0.756 | 0.693 | 1.134 | 0.811 | 0.830 | 0.699 |
| | FL = 24 hours | 0.781 | 0.711 | 1.217 | 0.855 | 1.220 | 0.855 | 0.787 | 0.712 | **0.775** | 0.715 | 1.246 | 0.864 | 0.787 | 0.712 | 1.213 | 0.853 | 0.924 | 0.761 |
| HL=4 hours | FL = 0.5 hour | 0.061 | 0.149 | 0.066 | 0.149 | 0.065 | 0.149 | 0.063 | 0.151 | 0.067 | 0.154 | 0.074 | 0.166 | **0.060** | **0.148** | 0.065 | 0.156 | 0.066 | 0.150 |
| | FL = 1 hour | 0.153 | **0.246** | 0.174 | 0.253 | 0.173 | 0.250 | 0.155 | 0.252 | 0.166 | 0.256 | 0.180 | 0.273 | **0.152** | 0.257 | 0.172 | 0.271 | 0.172 | 0.274 |
| | FL = 2 hours | **0.322** | **0.390** | 0.385 | 0.405 | 0.390 | 0.404 | 0.327 | 0.392 | 0.350 | 0.414 | 0.398 | 0.419 | 0.325 | 0.406 | 0.369 | 0.408 | 0.349 | 0.404 |
| | FL = 4 hours | 0.524 | 0.532 | 0.672 | 0.573 | 0.668 | 0.567 | 0.539 | 0.536 | 0.541 | 0.547 | 0.693 | 0.586 | **0.521** | **0.524** | 0.662 | 0.574 | 0.568 | 0.532 |
| | FL = 8 hours | **0.676** | 0.636 | 0.895 | 0.699 | 0.897 | 0.698 | 0.703 | 0.641 | 0.682 | 0.640 | 0.926 | 0.715 | 0.683 | 0.645 | 0.892 | 0.701 | 0.728 | **0.632** |
| | FL = 16 hours | 0.746 | 0.688 | 1.014 | 0.774 | 1.022 | 0.775 | 0.765 | 0.693 | **0.742** | 0.689 | 1.067 | 0.791 | 0.752 | 0.691 | 1.002 | 0.769 | 0.871 | 0.716 |
| | FL = 24 hours | **0.780** | **0.708** | 1.072 | 0.809 | 1.081 | 0.812 | 0.791 | 0.712 | 0.786 | 0.712 | 1.131 | 0.831 | 0.782 | 0.710 | 1.070 | 0.809 | 0.874 | 0.734 |
| HL=8 hours | FL = 0.5 hour | 0.062 | 0.150 | 0.065 | **0.149** | 0.066 | 0.152 | 0.064 | 0.152 | 0.067 | 0.153 | 0.076 | 0.177 | 0.063 | 0.153 | 0.065 | 0.158 | **0.061** | 0.151 |
| | FL = 1 hour | 0.151 | **0.249** | 0.167 | 0.252 | 0.171 | 0.253 | 0.159 | 0.253 | 0.165 | 0.256 | 0.193 | 0.284 | **0.150** | 0.252 | 0.166 | 0.257 | 0.157 | 0.259 |
| | FL = 2 hours | **0.326** | **0.392** | 0.367 | 0.399 | 0.374 | 0.401 | 0.340 | 0.414 | 0.351 | 0.415 | 0.398 | 0.432 | 0.331 | 0.407 | 0.352 | 0.400 | 0.334 | 0.395 |
| | FL = 4 hours | **0.527** | **0.536** | 0.612 | 0.556 | 0.625 | 0.558 | 0.528 | 0.541 | 0.544 | 0.557 | 0.695 | 0.601 | 0.532 | 0.544 | 0.609 | 0.557 | 0.636 | 0.569 |
| | FL = 8 hours | 0.733 | 0.653 | 0.796 | 0.671 | 0.811 | 0.672 | 0.678 | 0.640 | **0.671** | **0.637** | 0.876 | 0.708 | 0.694 | 0.654 | 0.784 | 0.670 | 0.797 | 0.653 |
| | FL = 16 hours | 0.770 | **0.690** | 0.914 | 0.745 | 0.931 | 0.749 | **0.751** | 0.697 | 0.778 | 0.698 | 0.985 | 0.772 | 0.768 | 0.692 | 0.899 | 0.740 | 0.842 | 0.724 |
| | FL = 24 hours | 0.782 | 0.710 | 0.951 | 0.766 | 0.972 | 0.773 | 0.788 | 0.711 | **0.776** | 0.716 | 1.044 | 0.799 | 0.793 | 0.713 | 0.947 | 0.771 | 0.889 | 0.744 |
| HL=24 hours | FL = 0.5 hour | **0.062** | 0.151 | 0.066 | 0.153 | 0.064 | 0.158 | 0.068 | 0.158 | 0.067 | 0.156 | 0.090 | 0.199 | 0.063 | 0.153 | 0.064 | 0.162 | 0.063 | 0.157 |
| | FL = 1 hour | **0.152** | 0.252 | 0.162 | 0.254 | 0.157 | **0.252** | 0.157 | 0.255 | 0.165 | 0.258 | 0.212 | 0.316 | 0.153 | 0.254 | 0.158 | 0.266 | 0.153 | 0.260 |
| | FL = 2 hours | **0.326** | 0.391 | 0.344 | 0.404 | 0.347 | 0.398 | 0.334 | 0.403 | 0.352 | 0.414 | 0.419 | 0.467 | 0.339 | 0.412 | 0.346 | 0.424 | 0.353 | 0.425 |
| | FL = 4 hours | **0.538** | 0.542 | 0.572 | 0.561 | 0.582 | 0.556 | 0.539 | 0.546 | 0.543 | 0.548 | 0.708 | 0.628 | 0.579 | 0.569 | 0.562 | 0.569 | 0.602 | 0.588 |
| | FL = 8 hours | 0.686 | **0.638** | 0.727 | 0.656 | 0.766 | 0.674 | 0.702 | 0.650 | **0.676** | 0.649 | 0.946 | 0.754 | 0.692 | 0.644 | 0.776 | 0.697 | 0.907 | 0.743 |
| | FL = 16 hours | 0.763 | 0.690 | 0.828 | 0.720 | 0.906 | 0.749 | 0.773 | 0.698 | **0.751** | **0.687** | 1.035 | 0.809 | 0.765 | 0.696 | 0.833 | 0.731 | 0.928 | 0.762 |
| | FL = 24 hours | **0.791** | 0.716 | 0.872 | 0.743 | 1.000 | 0.796 | 0.799 | 0.717 | 0.800 | 0.717 | 1.065 | 0.824 | 0.819 | 0.721 | 0.862 | 0.744 | 1.030 | 0.810 |
| HL=2 days | FL = 0.5 hour | **0.064** | 0.156 | 0.065 | 0.159 | 0.064 | 0.163 | 0.069 | 0.163 | 0.070 | 0.160 | 0.107 | 0.232 | 0.065 | 0.159 | 0.065 | 0.159 | **0.064** | **0.143** |
| | FL = 1 hour | 0.158 | **0.259** | 0.165 | 0.264 | 0.163 | 0.257 | 0.161 | 0.262 | 0.167 | 0.261 | 0.245 | 0.353 | **0.156** | 0.261 | 0.162 | 0.271 | 0.161 | 0.263 |
| | FL = 2 hours | **0.338** | **0.404** | 0.355 | 0.410 | 0.350 | 0.407 | 0.340 | 0.409 | 0.351 | 0.411 | 0.470 | 0.504 | 0.341 | 0.406 | 0.340 | 0.415 | 0.358 | 0.421 |
| | FL = 4 hours | 0.546 | 0.551 | 0.593 | 0.569 | 0.581 | 0.557 | **0.541** | 0.552 | 0.543 | 0.554 | 0.755 | 0.661 | 0.603 | 0.570 | 0.543 | **0.545** | 0.587 | 0.570 |
| | FL = 8 hours | 0.723 | 0.656 | 0.786 | 0.676 | 0.810 | 0.696 | **0.700** | 0.657 | **0.677** | **0.644** | 1.009 | 0.779 | 0.732 | 0.664 | 0.753 | 0.686 | 0.812 | 0.697 |
| | FL = 16 hours | **0.773** | **0.703** | 0.830 | 0.714 | 1.035 | 0.799 | 0.781 | 0.705 | 0.838 | 0.713 | 1.117 | 0.830 | 0.784 | 0.704 | 0.921 | 0.760 | 0.979 | 0.792 |
| | FL = 24 hours | **0.788** | **0.716** | 0.937 | 0.769 | 1.066 | 0.817 | 0.792 | 0.717 | 0.796 | 0.725 | 1.132 | 0.840 | 0.841 | 0.732 | 0.977 | 0.789 | 1.059 | 0.817 |
| HL=6 days | FL = 0.5 hour | 0.064 | 0.159 | 0.069 | 0.169 | 0.068 | 0.170 | 0.070 | 0.167 | 0.071 | 0.163 | 0.140 | 0.276 | 0.063 | **0.149** | 0.066 | 0.161 | 0.065 | 0.158 |
| | FL = 1 hour | **0.163** | **0.264** | 0.170 | 0.270 | 0.173 | 0.283 | 0.170 | 0.271 | 0.170 | 0.265 | 0.310 | 0.413 | 0.164 | 0.267 | 0.167 | 0.291 | 0.164 | 0.268 |
| | FL = 2 hours | **0.336** | **0.413** | 0.358 | 0.418 | 0.396 | 0.453 | 0.345 | 0.420 | 0.354 | 0.414 | 0.499 | 0.538 | 0.355 | 0.430 | 0.339 | 0.414 | 0.382 | 0.438 |
| | FL = 4 hours | 0.576 | 0.570 | 0.587 | 0.568 | 0.637 | 0.601 | **0.543** | 0.556 | 0.548 | **0.556** | 0.765 | 0.689 | 0.605 | 0.591 | 0.554 | 0.566 | 0.617 | 0.586 |
| | FL = 8 hours | **0.705** | **0.658** | 0.766 | 0.668 | 0.787 | 0.695 | 0.706 | 0.667 | 0.708 | 0.663 | 0.966 | 0.786 | 0.720 | 0.662 | 0.817 | 0.696 | 1.004 | 0.791 |
| | FL = 16 hours | 0.771 | 0.703 | 0.885 | 0.731 | 0.862 | 0.737 | 0.772 | 0.707 | **0.757** | **0.694** | 0.945 | 0.777 | 0.852 | 0.735 | 1.103 | 0.784 | 1.079 | 0.820 |
| | FL = 24 hours | **0.802** | 0.724 | 0.920 | 0.743 | 0.880 | 0.750 | 0.805 | 0.728 | 0.805 | **0.715** | 0.818 | 0.726 | 0.837 | 0.733 | 1.064 | 0.809 | 1.201 | 0.877 |

*Table 5.* Multi-Scale time series forecasting results across the DCLP5 dataset. The best is in **bold** and second-best is underlined. FL refers to forecasting length, HL refers to historical length.

| | Model | DSENet (Ours) | | TimeFilter (ICML, 2025) | | xPatch (AAAI, 2025) | | SST (CIKM, 2025) | | DLinear (AAAI, 2023) | | iTransformer (ICLR, 2024) | | Crossformer (ICLR, 2023) | | PatchTST (ICLR, 2023) | | NsT (NeurIPS, 2022) | |
|---|---|---|---|---|---|---|---|---|---|---|---|---|---|---|---|---|---|---|---|
| | Metrics | MSE | MAE | MSE | MAE | MSE | MAE | MSE | MAE | MSE | MAE | MSE | MAE | MSE | MAE | MSE | MAE | MSE | MAE |
| HL=1 hour | FL = 0.5 hour | **0.073** | **0.187** | 0.080 | 0.189 | 0.079 | **0.187** | 0.106 | 0.241 | 0.087 | 0.208 | 0.079 | 0.188 | 0.076 | 0.189 | 0.083 | 0.196 | 0.076 | 0.191 |
| | FL = 1 hour | **0.179** | **0.298** | 0.212 | 0.318 | 0.205 | 0.309 | 0.232 | 0.358 | 0.210 | 0.329 | 0.207 | 0.314 | 0.185 | 0.302 | 0.214 | 0.322 | 0.195 | 0.306 |
| | FL = 2 hours | **0.390** | **0.455** | 0.518 | 0.512 | 0.517 | 0.508 | 0.422 | 0.482 | 0.437 | 0.489 | 0.515 | 0.510 | 0.449 | 0.516 | 0.516 | 0.510 | 0.449 | 0.484 |
| | FL = 4 hours | 0.697 | **0.634** | 1.045 | 0.754 | 1.028 | 0.743 | 0.714 | 0.647 | 0.719 | 0.646 | 1.050 | 0.754 | **0.683** | 0.667 | 1.031 | 0.751 | 0.772 | 0.662 |
| | FL = 8 hours | **0.913** | **0.747** | 1.478 | 0.933 | 1.476 | 0.930 | 0.924 | 0.760 | 0.920 | 0.753 | 1.483 | 0.932 | 0.917 | 0.752 | 1.491 | 0.935 | 1.083 | 0.817 |
| | FL = 16 hours | 0.999 | 0.798 | 1.642 | 1.003 | 1.649 | 1.003 | 1.001 | 0.799 | 1.000 | 0.798 | 1.663 | 1.009 | 0.996 | 0.802 | 1.657 | 1.008 | 1.207 | 0.870 |
| | FL = 24 hours | 1.018 | 0.808 | 1.671 | 1.016 | 1.670 | 1.014 | 1.021 | 0.811 | 1.018 | 0.811 | 1.674 | 1.017 | 1.015 | 0.815 | 1.688 | 1.021 | 1.155 | 0.855 |
| HL=2 hours | FL = 0.5 hour | **0.074** | 0.187 | 0.081 | 0.194 | 0.078 | **0.185** | 0.090 | 0.216 | 0.083 | 0.200 | 0.083 | 0.197 | 0.078 | 0.198 | 0.084 | 0.193 | 0.076 | 0.192 |
| | FL = 1 hour | **0.179** | **0.298** | 0.225 | 0.332 | 0.207 | 0.312 | 0.209 | 0.337 | 0.208 | 0.326 | 0.221 | 0.328 | 0.182 | 0.305 | 0.219 | 0.324 | 0.184 | 0.301 |
| | FL = 2 hours | 0.403 | **0.460** | 0.536 | 0.526 | 0.517 | 0.514 | 0.440 | 0.495 | 0.443 | 0.491 | 0.518 | 0.514 | **0.401** | 0.467 | 0.532 | 0.522 | 0.441 | 0.485 |
| | FL = 4 hours | **0.718** | **0.644** | 1.016 | 0.748 | 0.997 | 0.735 | 0.721 | 0.646 | 0.732 | 0.651 | 1.028 | 0.750 | 0.729 | 0.653 | 1.004 | 0.744 | 0.808 | 0.672 |
| | FL = 8 hours | **0.926** | **0.754** | 1.391 | 0.909 | 1.395 | 0.904 | 0.934 | 0.772 | 0.933 | 0.757 | 1.445 | 0.925 | 0.943 | 0.759 | 1.395 | 0.907 | 1.111 | 0.820 |
| | FL = 16 hours | **0.999** | **0.803** | 1.532 | 0.970 | 1.545 | 0.970 | 1.021 | 0.812 | 1.010 | 0.804 | 1.552 | 0.976 | 1.011 | 0.809 | 1.539 | 0.971 | 1.106 | 0.842 |
| | FL = 24 hours | 1.013 | **0.810** | 1.545 | 0.976 | 1.552 | 0.976 | 1.023 | 0.812 | 1.025 | 0.811 | 1.557 | 0.980 | 1.021 | 0.814 | 1.548 | 0.977 | 1.130 | 0.848 |
| HL=4 hours | FL = 0.5 hour | **0.078** | 0.199 | 0.087 | 0.206 | 0.083 | 0.196 | 0.090 | 0.214 | 0.086 | 0.207 | 0.089 | 0.208 | 0.082 | 0.204 | 0.084 | 0.202 | 0.080 | 0.198 |
| | FL = 1 hour | 0.204 | **0.323** | 0.238 | 0.346 | 0.208 | 0.324 | 0.207 | 0.331 | 0.212 | 0.331 | 0.219 | 0.329 | **0.202** | 0.325 | 0.226 | 0.339 | 0.222 | 0.348 |
| | FL = 2 hours | 0.418 | **0.476** | 0.532 | 0.531 | 0.493 | 0.505 | 0.458 | 0.506 | 0.460 | 0.501 | 0.547 | 0.534 | 0.438 | 0.494 | 0.513 | 0.520 | 0.474 | 0.512 |
| | FL = 4 hours | 0.750 | **0.660** | 0.952 | 0.727 | 0.931 | 0.713 | 0.781 | 0.685 | 0.751 | 0.662 | 0.943 | 0.728 | 0.754 | 0.664 | 0.947 | 0.736 | 0.827 | 0.699 |
| | FL = 8 hours | **0.934** | **0.760** | 1.234 | 0.859 | 1.227 | 0.852 | 0.955 | 0.772 | 0.938 | 0.761 | 1.262 | 0.870 | 0.936 | 0.765 | 1.230 | 0.857 | 1.038 | 0.809 |
| | FL = 16 hours | 1.026 | 0.813 | 1.346 | 0.911 | 1.373 | 0.915 | 1.032 | 0.814 | 1.017 | **0.806** | 1.387 | 0.930 | 1.040 | 0.822 | 1.357 | 0.915 | 1.084 | 0.836 |
| | FL = 24 hours | **1.022** | 0.815 | 1.365 | 0.917 | 1.361 | 0.915 | 1.033 | 0.817 | 1.024 | 0.821 | 1.375 | 0.923 | 1.031 | **0.814** | 1.364 | 0.920 | 1.072 | 0.835 |
| HL=8 hours | FL = 0.5 hour | 0.083 | 0.205 | 0.082 | 0.200 | 0.081 | 0.196 | 0.093 | 0.221 | 0.085 | 0.205 | 0.119 | 0.253 | **0.080** | 0.200 | 0.092 | 0.218 | 0.100 | 0.229 |
| | FL = 1 hour | **0.199** | **0.323** | 0.206 | 0.324 | 0.207 | 0.324 | 0.211 | 0.337 | 0.213 | 0.333 | 0.247 | 0.362 | 0.206 | 0.329 | 0.240 | 0.361 | 0.233 | 0.361 |
| | FL = 2 hours | **0.433** | **0.490** | 0.524 | 0.532 | 0.457 | 0.491 | 0.447 | 0.498 | 0.456 | 0.503 | 0.566 | 0.558 | 0.469 | 0.525 | 0.491 | 0.517 | 0.561 | 0.547 |
| | FL = 4 hours | 0.756 | 0.668 | 0.862 | 0.706 | 0.822 | 0.677 | 0.772 | 0.679 | **0.745** | 0.683 | 0.945 | 0.733 | 0.758 | **0.666** | 0.814 | 0.687 | 0.779 | 0.677 |
| | FL = 8 hours | **0.931** | **0.763** | 1.078 | 0.810 | 1.080 | 0.803 | 0.945 | 0.771 | 0.933 | 0.769 | 1.182 | 0.850 | 0.944 | 0.773 | 1.085 | 0.815 | 1.048 | 0.812 |
| | FL = 16 hours | 1.007 | **0.807** | 1.181 | 0.862 | 1.198 | 0.859 | 1.018 | 0.812 | 1.007 | 0.811 | 1.274 | 0.892 | 1.011 | 0.815 | 1.162 | 0.854 | 1.114 | 0.845 |
| | FL = 24 hours | **1.003** | **0.805** | 1.181 | 0.859 | 1.197 | 0.859 | 1.019 | 0.813 | 1.007 | 0.806 | 1.267 | 0.888 | 1.017 | 0.809 | 1.202 | 0.869 | 1.138 | 0.863 |
| HL=24 hours | FL = 0.5 hour | 0.083 | 0.203 | 0.087 | 0.211 | **0.080** | **0.199** | 0.087 | 0.212 | 0.087 | 0.208 | 0.157 | 0.298 | 0.082 | 0.204 | 0.088 | 0.216 | 0.084 | 0.207 |
| | FL = 1 hour | 0.208 | 0.331 | 0.221 | 0.338 | 0.209 | **0.326** | 0.212 | 0.339 | 0.210 | 0.332 | 0.360 | 0.455 | 0.210 | 0.338 | **0.206** | 0.333 | 0.243 | 0.381 |
| | FL = 2 hours | 0.444 | **0.497** | 0.481 | 0.516 | 0.460 | 0.504 | 0.456 | 0.508 | 0.461 | 0.509 | 0.707 | 0.635 | **0.432** | 0.493 | 0.441 | 0.493 | 0.505 | 0.532 |
| | FL = 4 hours | 0.746 | 0.673 | 0.772 | 0.666 | 0.738 | 0.648 | 0.751 | 0.674 | 0.724 | 0.656 | 1.045 | 0.786 | **0.718** | 0.664 | 0.831 | 0.689 | 0.858 | 0.717 |
| | FL = 8 hours | **0.905** | **0.756** | 0.947 | 0.763 | 0.972 | 0.771 | 0.914 | 0.761 | 0.907 | 0.758 | 1.179 | 0.856 | 0.909 | 0.760 | 0.980 | 0.787 | 1.190 | 0.856 |
| | FL = 16 hours | 0.965 | 0.786 | 1.021 | 0.799 | 1.127 | 0.828 | 0.968 | 0.792 | **0.955** | 0.790 | 1.285 | 0.896 | 1.005 | 0.799 | 1.056 | 0.815 | 1.159 | 0.860 |
| | FL = 24 hours | **0.977** | **0.793** | 1.060 | 0.816 | 1.159 | 0.844 | 0.995 | 0.804 | 0.980 | 0.795 | 1.233 | 0.882 | 1.010 | 0.802 | 1.190 | 0.866 | 1.556 | 0.994 |
| HL=2 days | FL = 0.5 hour | 0.092 | 0.218 | 0.097 | 0.225 | 0.090 | 0.219 | 0.093 | 0.221 | **0.089** | **0.213** | 0.246 | 0.376 | 0.095 | 0.226 | 0.096 | 0.224 | 0.101 | 0.231 |
| | FL = 1 hour | 0.230 | 0.350 | 0.232 | 0.351 | 0.224 | 0.350 | 0.222 | 0.350 | 0.242 | **0.342** | 0.446 | 0.504 | **0.217** | 0.346 | 0.232 | 0.359 | 0.226 | 0.351 |
| | FL = 2 hours | **0.459** | **0.513** | 0.481 | 0.514 | 0.489 | 0.518 | 0.476 | 0.515 | 0.479 | 0.514 | 0.808 | 0.686 | 0.462 | 0.516 | 0.478 | 0.516 | 0.551 | 0.547 |
| | FL = 4 hours | 0.767 | 0.677 | 0.934 | 0.743 | 0.787 | 0.671 | **0.767** | 0.686 | 0.769 | **0.665** | 1.172 | 0.835 | 0.769 | 0.678 | 0.789 | 0.671 | 0.856 | 0.714 |
| | FL = 8 hours | **0.952** | **0.770** | 1.057 | 0.800 | 1.094 | 0.814 | 0.972 | 0.779 | 0.963 | 0.780 | 1.350 | 0.909 | 0.981 | 0.771 | 1.091 | 0.818 | 1.033 | 0.800 |
| | FL = 16 hours | **1.028** | **0.806** | 1.186 | 0.862 | 1.156 | 0.851 | 1.040 | 0.812 | 1.038 | 0.813 | 1.367 | 0.921 | 1.058 | 0.809 | 1.068 | 0.816 | 1.144 | 0.838 |
| | FL = 24 hours | **1.027** | **0.811** | 1.227 | 0.874 | 1.258 | 0.890 | 1.067 | 0.826 | 1.037 | 0.818 | 1.316 | 0.910 | 1.156 | 0.850 | 1.065 | 0.813 | 1.201 | 0.878 |
| HL=6 days | FL = 0.5 hour | **0.092** | 0.219 | 0.129 | 0.270 | 0.114 | 0.255 | 0.119 | 0.258 | 0.099 | 0.226 | 0.363 | 0.472 | 0.095 | 0.229 | 0.095 | 0.225 | 0.109 | 0.238 |
| | FL = 1 hour | **0.229** | **0.358** | 0.305 | 0.416 | 0.255 | 0.382 | 0.269 | 0.397 | 0.244 | 0.367 | 0.639 | 0.614 | 0.239 | 0.375 | 0.257 | 0.376 | 0.290 | 0.408 |
| | FL = 2 hours | **0.495** | 0.540 | 0.680 | 0.621 | 0.618 | 0.594 | 0.657 | 0.637 | 0.497 | **0.527** | 0.845 | 0.719 | 0.560 | 0.568 | 0.636 | 0.612 | 0.658 | 0.620 |
| | FL = 4 hours | 0.866 | 0.718 | 1.057 | 0.786 | 1.024 | 0.783 | 0.938 | 0.774 | 0.889 | 0.781 | 1.084 | 0.822 | **0.759** | **0.666** | 0.933 | 0.737 | 1.114 | 0.827 |
| | FL = 8 hours | 0.966 | **0.778** | 1.419 | 0.928 | 1.133 | 0.836 | 1.011 | 0.814 | **0.957** | 0.794 | 1.166 | 0.858 | 1.001 | 0.788 | 1.153 | 0.846 | 1.124 | 0.857 |
| | FL = 16 hours | 1.015 | 0.816 | 1.545 | 0.976 | 1.187 | 0.864 | 1.064 | 0.817 | 1.064 | 0.817 | 1.218 | 0.878 | 1.050 | 0.833 | 1.210 | 0.884 | 1.092 | 0.840 |
| | FL = 24 hours | 1.023 | **0.814** | 1.637 | 1.007 | 1.193 | 0.868 | 1.045 | 0.820 | 1.077 | 0.825 | 1.253 | 0.892 | 1.025 | 0.823 | 1.244 | 0.894 | 1.418 | 0.933 |

**Lemma C.8.** *Assume that there exist constants* $0 < q_{\text{hi}} \leq q_{\text{lo}}$, *let* $|Q(\omega)| \leq q_{\text{hi}}$, $|\omega| > \omega_c$ *and* $|Q(\omega)| \geq q_{\text{lo}}, |\omega| \leq \omega_c$. *Then* $\tilde{\varepsilon}(\omega_c) \leq \varepsilon(\omega_c)$

*Proof.* We denote the high-frequency energy is $A = \int_{|\omega|>\omega_c} |H(\omega)|^2 \, d\omega$ and the low-frequency energy is $B = \int_{|\omega|<\omega_c} |H(\omega)|^2 \, d\omega$ The original leakage is $\varepsilon = \frac{A}{A+B}$. Under the new operator, the high-frequency turns to $\tilde{A} = \int_{|\omega|>\omega_c} |Q(\omega)|^2 |H(\omega)|^2 \, d\omega \leq q_{\text{hi}}^2 A$ and the low-frequency energy turns to $\tilde{B} = \int_{|\omega|<\omega_c} |Q(\omega)|^2 |H(\omega)|^2 \, d\omega \geq q_{\text{lo}}^2 B$. Thus the total energy is $\tilde{E} = \tilde{A} + \tilde{B} \geq \tilde{A} + q_{\text{lo}}^2 B$.

As a result, the high-frequency leakage now is $\tilde{\varepsilon} = \frac{\tilde{A}}{\tilde{E}} \leq \frac{\tilde{A}}{\tilde{A}+q_{\text{lo}}^2 B} \leq \frac{q_{\text{hi}}^2 A}{q_{\text{hi}}^2 A + q_{\text{lo}}^2 B}$. Notice that $q_{\text{lo}}^2 \geq q_{\text{hi}}^2$, so $q_{\text{hi}}^2 A + q_{\text{lo}}^2 B \geq q_{\text{hi}}^2 A + q_{\text{hi}}^2 B = q_{\text{hi}}^2 (A + B)$. Therefore, it has:

$$\tilde{\varepsilon} \leq \frac{q_{\text{hi}}^2 A}{q_{\text{hi}}^2 (A + B)} = \frac{A}{A + B} = \varepsilon \tag{38}$$

Such a formulation mimics the behavior of a low-pass filter: intuitively, it suppresses high-frequency components while preserving low-frequency ones, thereby rendering the overall system response more strongly low-pass.

Specifically, for Gate-Mamba, it can be viewed as a forget gate embedded in a bi-directional structure. For the bi-directional structure, the frequency response of forward pass is $Y_f(\omega) = H_f(\omega) X(\omega)$ and the backward pass is $Y_b(\omega) = H_f(-\omega) X(\omega)$. After that we use DSSAF Alg. to combine the two paths, which can be denoted as $\tilde{H}_g(\omega) = \alpha H_f(\omega) + \beta H_f(-\omega) = Q_g(\omega) H_{g,0}(\omega)$. Rewrite this formula to the following type:

$$H_{g,0}(\omega) = H_f(\omega), \qquad Q_g(\omega) = \alpha + \beta \frac{H_f(-\omega)}{H_f(\omega)} \tag{39}$$

In other words, the role of bidirectionality combined with the fusion gating is to introduce a frequency-dependent reweighting.

**Assumption C.9.** Consider the Gate-Mamba block under the local linearization in Assumption C.1, which induces an *effective* frequency response $H_f(\omega; x)$ around an input $x \in \mathcal{M}$. In the low-frequency regime and for long-memory settings (where the gating varies slowly within a local temporal neighborhood), we assume that the effective response approximately satisfies $H_f(-\omega; x) \approx \overline{H_f(\omega; x)}, |\omega| \leq \omega_c$.

In the Assumption C.9, in the low-frequency domain, the phases are nearly conjugate, leading to constructive in-phase reinforcement upon aggregation. In contrast, in the high-frequency band, the phase discrepancy becomes pronounced, causing responses from opposite directions to partially cancel out. Under this, for low-frequency, $|Q_g(\omega)| = \left| \alpha + \beta \frac{H_f(-\omega)}{H_f(\omega)} \right| \approx |\alpha + \beta| = 1$, $|\omega| < \omega_c$. For high-frequency, $|Q_g(\omega)| = |\alpha + \beta e^{j\Delta\phi(\omega)}| \leq q_{\text{hi}} < 1$, $|\omega| > \omega_c$. As a result, we can get the same structure in Lemma C.8, that is, $|Q_g(\omega)| \geq q_{\text{lo}} \approx 1$, $|\omega| < \omega_c$ and $|Q_g(\omega)| \leq q_{\text{hi}} < 1$, $|\omega| > \omega_c$. Thus:

$$\varepsilon_g^{(\text{Bi-Mamba})}(\omega_c) \leq \varepsilon_g^{(\text{Vanilla})}(\omega_c) \tag{40}$$

The introduction of the forget gate is equivalent to multiplying the left- and right-branch outputs of Mamba by two slowly varying weighting functions. As a result, the resulting output admits the same structural form as that in Lemma C.8, which leads to the following conclusion:

$$\varepsilon_g^{(\text{DFB-Mamba})}(\omega_c) \leq \varepsilon_g^{(\text{Bi-Mamba})}(\omega_c) \leq \varepsilon_g^{(\text{Vanilla})}(\omega_c) \tag{41}$$

***For the local stream,*** there will be anther version of Lemma which is similar to Lemma C.8 with the opposite inequality symbol. The derivation for LoRE is analogous to that of DFB-Mamba and has already been rigorously analyzed in Section B, where we show that the LoRE block enhances local sensitivity. We therefore omit the detailed derivation here. Notably, the structure of LoRE allows it to selectively amplify responses along a small number of feature directions while leaving the remaining directions nearly unchanged. If these selected directions align with high-frequency perturbations during training, LoRE exhibits larger gains in the high-frequency band while remaining close to unity in the low-frequency band. Consequently, the resulting formulation can still be cast into a form analogous to that in Lemma C.8, leading directly to the following conclusion:

$$\varepsilon_l^{(\text{LoRE-LWSA})}(\omega_c) \leq \varepsilon_l^{(\text{Vanilla})}(\omega_c) \tag{42}$$

*Table 6.* Inter-stream CKA analysis between the global and local streams before feature fusion. Lower CKA values indicate weaker inter-stream coupling and better stream decoupling. ΔCKA denotes the relative change compared with the baseline.

| Model | Overall CKA | | Batch-wise CKA | | | | |
|---|---|---|---|---|---|---|---|
| | Inter-stream CKA ↓ | ΔCKA | Mean CKA ↓ | ΔCKA | Median CKA ↓ | Q75 ↓ | CKA Std |
| Baseline | 0.567 | – | 0.874 | – | 0.907 | 0.974 | 0.116 |
| Baseline + DFB-Mamba | 0.560 | -1.235% | 0.869 | -0.572% | 0.884 | 0.940 | **0.092** |
| Baseline + LoRE | 0.524 | -7.584% | 0.598 | -31.579% | 0.660 | 0.768 | 0.237 |
| **Ours** | **0.472** | **-16.755%** | **0.186** | **-78.719%** | **0.159** | **0.357** | 0.227 |

Now we look back to Equation (35), the DFB-Mamba and LoRE lower the $\varepsilon_g$ and $\varepsilon_l$, thus $D_{\text{spec}}$ will get larger. In other words, the simultaneous presence of the two components indeed enlarges the discrepancy between the two streams, thereby forcing the global and local streams to learn more complementary representations, while simultaneously enhancing the model's capability to capture both global characteristics and local fluctuations. As a result, we successfully proof the effectiveness of proposed Dual-Stream Enhancement Mechanism!

We further perform an inter-stream CKA analysis to quantitatively examine the representation discrepancy between the global and local streams. The CKA similarity is computed between the two streams before feature fusion, where a lower value indicates weaker inter-stream coupling and thus a stronger decoupling effect. We first report the average inter-stream CKA as an overall measure of stream coupling. Nevertheless, relying only on a single averaged CKA value may hide batch-level variations. To provide a more comprehensive evaluation, we additionally report batch-wise CKA statistics, including the mean, median, and upper quartile (Q75). The mean reflects the overall coupling tendency, the median measures the typical inter-stream similarity, and Q75 characterizes relatively high-coupling cases across batches. As reported in Table 6, the proposed dual-stream enhancement consistently reduces all CKA statistics. This demonstrates that our method not only lowers the average coupling between global and local streams, but also suppresses high-coupling cases at the batch level, thereby promoting more distinguishable and complementary stream-specific representations.

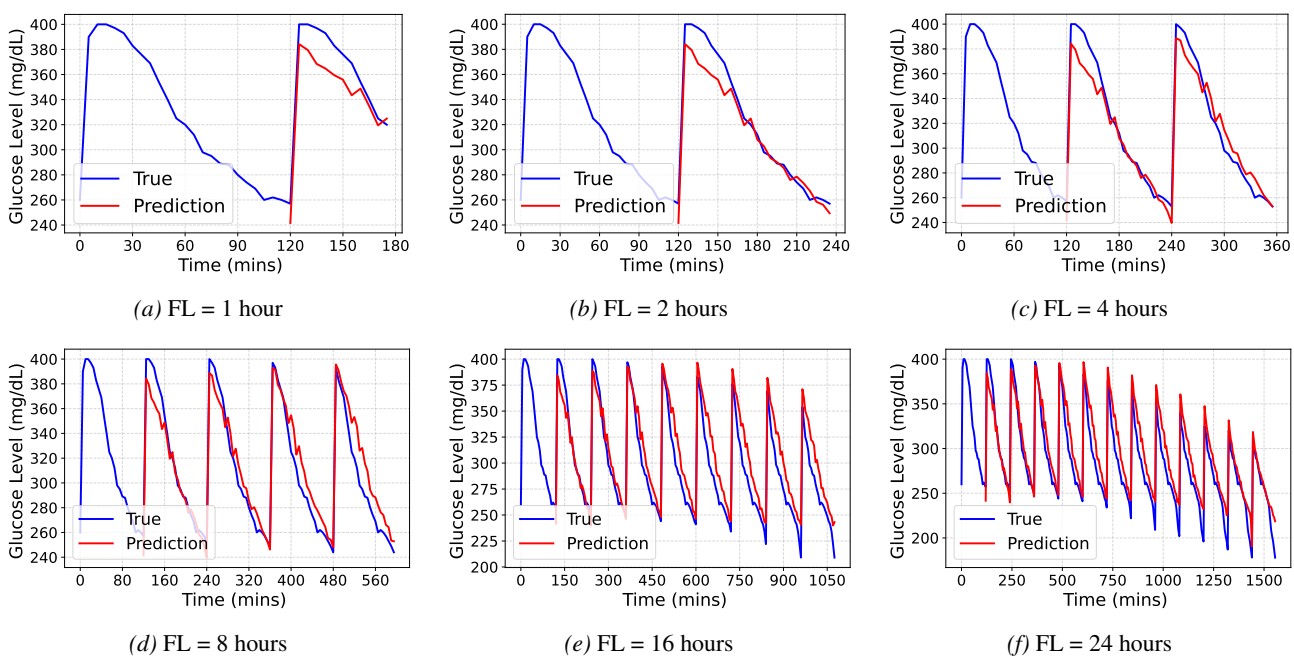

*Figure 14.* Visualization results of DCLP5 dataset of DSENet.

# D. Forecasting and Visualization Results of DCLP5 Dataset

The complete experimental results on both public dataset for blood glucose prediction are reported in Table 4 and Table 5. It can be observed that our model consistently achieves the best performance across multi-time scales, thereby further

*Table 7.* Zero-shot prediction results under different forecasting lengths (FL). The best is in **bold** and second-best is underlined.

| Model | 0.5 hour | | 1 hour | | 2 hours | | 4 hours | | 8 hours | | 16 hours | | 24 hours | | 3 days | |
|---|---|---|---|---|---|---|---|---|---|---|---|---|---|---|---|---|
| | MSE | MAE | MSE | MAE | MSE | MAE | MSE | MAE | MSE | MAE | MSE | MAE | MSE | MAE | MSE | MAE |
| **Ours** | 0.060 | 0.146 | **0.133** | **0.239** | 0.311 | **0.387** | **0.538** | **0.542** | 0.692 | 0.648 | **0.771** | **0.688** | 0.820 | **0.712** | **0.878** | **0.739** |
| TimeFilter | 0.051 | 0.146 | 0.135 | 0.245 | 0.319 | 0.393 | 0.564 | 0.550 | 0.741 | 0.656 | 0.825 | 0.705 | 0.848 | 0.723 | 0.924 | 0.759 |
| xPatch | 0.163 | 0.305 | 0.265 | 0.385 | 0.447 | 0.497 | 0.592 | 0.578 | 0.702 | **0.639** | 0.861 | 0.742 | 0.868 | 0.757 | 0.892 | 0.804 |
| SST | 0.071 | 0.167 | 0.170 | 0.271 | 0.314 | 0.429 | 0.543 | 0.545 | 0.720 | 0.657 | 0.801 | 0.698 | **0.815** | 0.714 | 1.247 | 0.851 |
| DLinear | **0.046** | **0.140** | 0.134 | 0.241 | 0.338 | 0.393 | 0.562 | 0.547 | **0.688** | 0.670 | 0.855 | 0.715 | 0.878 | 0.729 | 0.928 | 0.760 |
| iTransformer | 0.124 | 0.258 | 0.269 | 0.375 | 0.484 | 0.518 | 0.707 | 0.635 | 0.920 | 0.738 | 0.948 | 0.755 | 0.984 | 0.773 | 1.147 | 0.840 |
| Crossformer | 0.051 | 0.150 | 0.136 | 0.252 | **0.309** | 0.388 | 0.542 | 0.580 | 0.749 | 0.652 | 0.852 | 0.712 | 0.917 | 0.743 | 0.956 | 0.764 |
| PatchTST | 0.054 | 0.156 | 0.140 | 0.252 | 0.312 | 0.396 | 0.549 | 0.553 | 0.758 | 0.660 | 0.943 | 0.749 | 0.978 | 0.758 | 1.125 | 0.832 |
| NsT | 0.056 | 0.153 | 0.152 | 0.250 | 0.391 | 0.428 | 0.606 | 0.564 | 0.918 | 0.736 | 0.911 | 0.743 | 0.985 | 0.773 | 1.093 | 0.815 |

validating its strong generalization capability. Moreover, these results further corroborate our conclusion that the proposed model can serve as a new baseline for non-stationary blood glucose time-series prediction. Following the OhioT1DM dataset, we also visualize the results on the DCLP5 dataset, where a 2-hour history window is used to predict future outcomes at multiple temporal scales as shown in Figure 14.

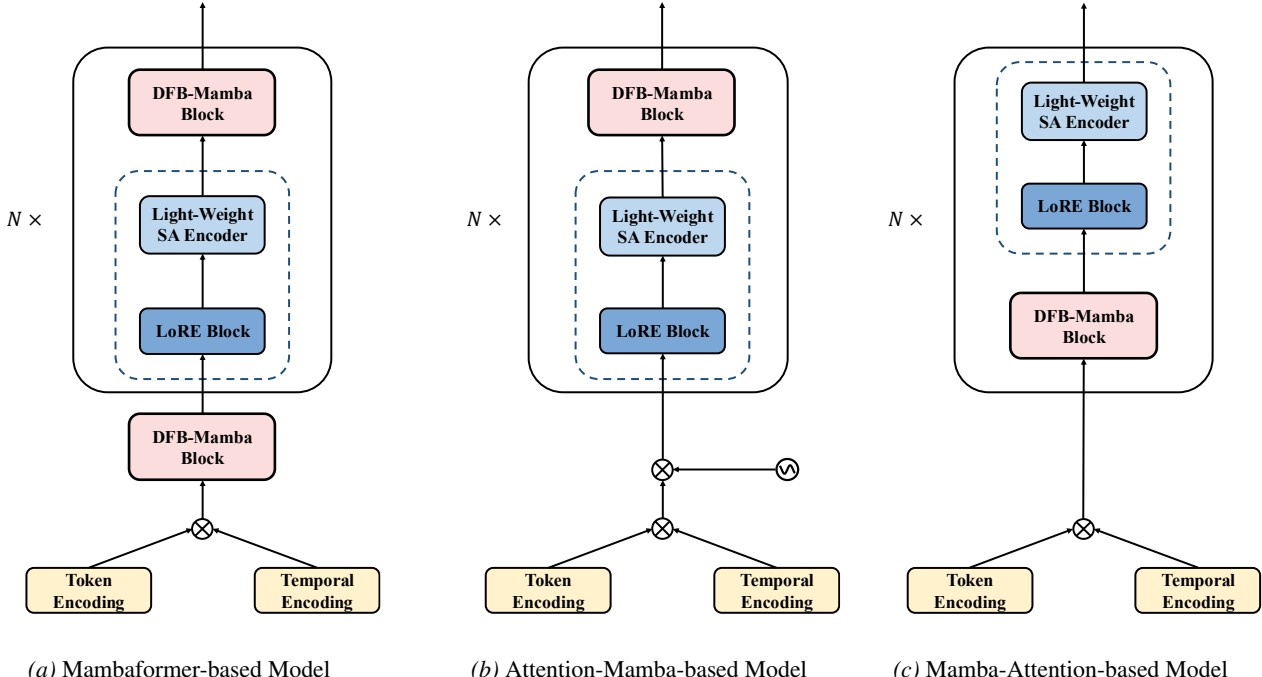

*(a)* Mambaformer-based Model          *(b)* Attention-Mamba-based Model          *(c)* Mamba-Attention-based Model

*Figure 15.* Three paradigms based on Mamba and Attention mechanism, which belonging to the MambaFormer family.

## E. Zero-Shot Prediction Results

In real-world scenarios, blood glucose dynamics exhibit substantial inter-individual variability, resulting in strong personalization effects. Most existing blood glucose prediction models are trained on fixed datasets and suffer from limited transferability, leading to degraded performance on unseen patients. To evaluate the generalization ability of our approach, we conduct experiments under a zero-shot forecasting setting, where the trained model with OhioT1DM 2018 is directly applied to newly released subjects from the OhioT1DM 2020 cohort without any fine-tuning. As reported in Table 7, our model demonstrates excellent transfer performance. To further assess long-term generalization, we introduce an additional prediction horizon of $F = 3 days$. Under this challenging setting, DSENet outperforms the second-best method by reducing MAE by 8.9%, indicating strong robustness and long-horizon forecasting capability.

## F. More Ablation Study of DSENet

To further verify the rationality and uniqueness of the DSENet architecture, that is the necessity of the dual-stream structure and the dual-stream enhancement mechanism. We introduce the structural paradigms defined by Mambaformer (Xu et al., 2024) to conduct more comprehensive ablation study. Specifically, we evaluate the effectiveness of serial architectures and, under the serial paradigm, investigate how different placements of the global and local streams affect performance. The structure of Mambaformer-style models is illustrated in Figure 15, where we replace the Vanilla Mamba and Vanilla Attention modules with the proposed DFB-Mamba and the attention mechanism augmented with LoRE blocks, respectively. To further verify the effectiveness of the dual-stream enhancement mechanism we proposed, we explicitly removed the LoRE block in the local stream, retained only the lightweight self-attention module, and used Vanilla Mamba to replace the DFB-Mamba we proposed in global stream. By this way of restoring feature coupling, we eliminate the dual-stream enhancement mechanism.

*Table 8.* More Ablation study on two datasets under three different historical lengths and three different forecasting lengths.

|  | OhioT1DM | | | DCLP5 | | |
|---|---|---|---|---|---|---|
|  | HL=1h | HL=2h | HL=4h | HL=1h | HL=2h | HL=4h |
| **Ours** | **0.389** | **0.394** | **0.390** | **0.455** | **0.460** | **0.476** |
| Mambaformer-based | 0.412 | 0.408 | 0.425 | 0.492 | 0.542 | 0.517 |
| Attention-Mamba-based | 0.447 | 0.407 | 0.425 | 0.469 | 0.469 | 0.512 |
| Mamba-Attention-based | 0.417 | 0.413 | 0.431 | 0.468 | 0.482 | 0.508 |
| w/o Dual-Stream Enhancement Mechanism | 0.397 | 0.400 | 0.392 | 0.482 | 0.495 | 0.506 |
|  | OhioT1DM | | | DCLP5 | | |
|  | FL=0.5h | FL=1h | FL=2h | FL=0.5h | FL=1h | FL=2h |
| **Ours** | **0.149** | **0.239** | **0.389** | **0.187** | **0.298** | **0.455** |
| Mambaformer-based | 0.170 | 0.258 | 0.404 | 0.191 | 0.304 | 0.461 |
| Attention-Mamba-based | 0.169 | 0.280 | 0.452 | 0.200 | 0.353 | 0.463 |
| Mamba-Attention-based | 0.156 | 0.266 | 0.417 | 0.191 | 0.308 | 0.466 |
| w/o Dual-Stream Enhancement Mechanism | 0.152 | 0.250 | 0.397 | 0.241 | 0.358 | 0.482 |

The ablation results reported in Table 8 demonstrate that the parallel dual-stream model constitutes the most effective architecture for time-series forecasting. Moreover, the proposed dual-stream enhancement mechanism substantially improves forecasting performance and strengthens the model's sensitivity to local abrupt variations.

*Table 9.* Robustness evaluation of DSENet under unseen input perturbations. $\Delta$MSE and $\Delta$MAE denote the relative performance changes compared with the clean setting.

| Perturbation | MSE | $\Delta$MSE | MAE | $\Delta$MAE |
|---|---|---|---|---|
| **Clean** | **0.516** | – | **0.523** | – |
| Gaussian, $\sigma = 0.01$ | 0.517 | 0.194% | 0.524 | 0.191% |
| Gaussian, $\sigma = 0.05$ | 0.522 | 1.163% | 0.529 | 1.147% |
| Spike, $r = 0.01$, $a = 0.1$ | 0.517 | 0.194% | 0.524 | 0.191% |
| Spike, $r = 0.05$, $a = 0.2$ | 0.520 | 0.775% | 0.527 | 0.765% |
| Step shift, $\delta = 0.1$ | 0.518 | 0.388% | 0.525 | 0.382% |
| Step shift, $\delta = 0.2$ | 0.518 | 0.388% | 0.527 | 0.765% |

## G. Robustness to Noise and Outliers

We further evaluate the robustness of DSENet under different input perturbations, since sensor noise and abrupt distributional variations are common in clinical forecasting scenarios. Specifically, we inject three types of unseen perturbations into the test inputs, including Gaussian noise, spike-like disturbances, and abrupt step shifts.

Before patch embedding, DSENet applies RevIN to reduce the influence of distribution drift in the input data. Moreover, DSENet does not directly amplify raw point-wise fluctuations. Instead, the local branch operates on patch-level representations, and the final prediction is jointly constrained by both the local and global streams. Therefore, the model is less likely to overfit or amplify isolated noisy observations. As shown in Table X, DSENet maintains stable forecasting

performance under all perturbation settings. Compared with the clean setting, the increases in MSE and MAE remain small even under stronger Gaussian noise, spike disturbances, and abrupt step shifts. These results indicate that DSENet exhibits good robustness to unseen noise and outlier-like perturbations.

*Table 10.* Multivariate time series forecasting results across five datasets. The historical length $HL = 96$ and the forecasting length $FL \in \{96, 192, 336, 720\}$. The best is in **bold** and second-best is underlined.

| Models | | DSENet (Ours) | | TimeFilter (ICML, 2025) | | MSGNet (AAAI, 2024) | | TimeMixer (ICLR, 2024) | | iTransformer (ICLR, 2024) | | MICN (ICLR, 2023) | | TimesNet (ICLR, 2023) | | PatchTST (ICLR, 2023) | | NsT (NeurIPS, 2022) | |
|---|---|---|---|---|---|---|---|---|---|---|---|---|---|---|---|---|---|---|---|
| Metrics | | MSE | MAE | MSE | MAE | MSE | MAE | MSE | MAE | MSE | MAE | MSE | MAE | MSE | MAE | MSE | MAE | MSE | MAE |
| ETTh1 | 96 | **0.370** | **0.396** | 0.373 | 0.398 | 0.390 | 0.411 | 0.375 | 0.400 | 0.386 | 0.405 | 0.421 | 0.431 | 0.384 | 0.402 | 0.370 | 0.400 | 0.513 | 0.491 |
| | 192 | **0.410** | 0.428 | 0.413 | **0.420** | 0.442 | 0.442 | 0.429 | 0.421 | 0.441 | 0.436 | 0.474 | 0.487 | 0.436 | 0.429 | 0.413 | 0.429 | 0.534 | 0.504 |
| | 336 | 0.458 | **0.440** | **0.454** | 0.444 | 0.480 | 0.468 | 0.484 | 0.458 | 0.487 | 0.458 | 0.569 | 0.551 | 0.491 | 0.469 | 0.474 | 0.451 | 0.588 | 0.535 |
| | 720 | 0.460 | **0.456** | **0.448** | 0.489 | 0.494 | 0.488 | 0.498 | 0.482 | 0.503 | 0.491 | 0.770 | 0.672 | 0.521 | 0.500 | **0.448** | 0.468 | 0.643 | 0.616 |
| ETTh2 | 96 | 0.293 | 0.348 | **0.288** | 0.339 | 0.328 | 0.371 | 0.294 | 0.343 | 0.297 | 0.349 | 0.299 | 0.364 | 0.340 | 0.374 | 0.302 | 0.348 | 0.476 | 0.458 |
| | 192 | 0.377 | 0.394 | **0.367** | 0.396 | 0.402 | 0.414 | 0.372 | **0.392** | 0.380 | 0.400 | 0.441 | 0.454 | 0.402 | 0.414 | 0.388 | 0.400 | 0.512 | 0.493 |
| | 336 | **0.389** | **0.415** | 0.407 | 0.430 | 0.435 | 0.443 | 0.442 | 0.442 | 0.428 | 0.432 | 0.454 | 0.654 | 0.452 | 0.452 | 0.426 | 0.433 | 0.552 | 0.551 |
| | 720 | **0.410** | **0.434** | 0.411 | 0.435 | 0.417 | 0.441 | 0.439 | 0.451 | 0.427 | 0.445 | 0.956 | 0.716 | 0.462 | 0.468 | 0.431 | 0.446 | 0.562 | 0.560 |
| ETTm1 | 96 | 0.318 | **0.353** | **0.313** | 0.354 | 0.319 | 0.366 | 0.320 | 0.357 | 0.334 | 0.368 | 0.328 | 0.362 | 0.338 | 0.375 | 0.329 | 0.367 | 0.386 | 0.398 |
| | 192 | 0.361 | **0.380** | **0.356** | 0.380 | 0.376 | 0.397 | 0.362 | 0.384 | 0.377 | 0.391 | 0.390 | 0.408 | 0.387 | 0.410 | 0.385 | 0.399 | 0.444 | 0.495 |
| | 336 | 0.396 | **0.402** | **0.386** | 0.404 | 0.417 | 0.422 | 0.404 | 0.411 | 0.426 | 0.420 | 0.408 | 0.426 | 0.410 | 0.411 | 0.399 | 0.410 | 0.495 | 0.464 |
| | 720 | 0.456 | **0.439** | 0.452 | 0.440 | 0.481 | 0.458 | 0.463 | 0.441 | 0.491 | 0.459 | 0.481 | 0.476 | 0.478 | 0.450 | 0.454 | 0.439 | 0.585 | 0.516 |
| ETTm2 | 96 | 0.172 | 0.253 | **0.169** | 0.255 | 0.177 | 0.262 | 0.175 | 0.258 | 0.180 | 0.264 | 0.179 | 0.275 | 0.187 | 0.267 | 0.175 | 0.259 | 0.192 | 0.274 |
| | 192 | 0.239 | **0.299** | **0.236** | 0.300 | 0.307 | 0.312 | 0.240 | 0.301 | 0.250 | 0.309 | 0.307 | 0.376 | 0.249 | 0.309 | 0.241 | 0.302 | 0.280 | 0.339 |
| | 336 | **0.298** | **0.340** | 0.300 | 0.341 | 0.312 | 0.346 | 0.306 | 0.343 | 0.311 | 0.348 | 0.325 | 0.388 | 0.321 | 0.351 | 0.305 | 0.343 | 0.334 | 0.361 |
| | 720 | 0.392 | **0.391** | **0.390** | 0.393 | 0.414 | 0.403 | 0.396 | 0.399 | 0.412 | 0.407 | 0.502 | 0.490 | 0.408 | 0.403 | 0.402 | 0.400 | 0.417 | 0.413 |
| Exchange | 96 | **0.083** | 0.208 | 0.084 | **0.202** | 0.103 | 0.229 | 0.087 | 0.205 | 0.086 | 0.206 | 0.102 | 0.235 | 0.107 | 0.234 | 0.088 | 0.205 | 0.111 | 0.237 |
| | 192 | **0.172** | **0.298** | 0.180 | 0.300 | 0.212 | 0.333 | 0.185 | 0.306 | 0.177 | 0.299 | 0.174 | 0.316 | 0.226 | 0.344 | 0.176 | 0.299 | 0.219 | 0.335 |
| | 336 | 0.308 | 0.400 | 0.339 | 0.419 | 0.399 | 0.461 | 0.369 | 0.443 | 0.331 | 0.417 | 0.318 | 0.442 | 0.367 | 0.448 | **0.301** | **0.397** | 0.421 | 0.476 |
| | 720 | **0.807** | **0.683** | 0.880 | 0.704 | 1.033 | 0.779 | 0.865 | 0.697 | 0.847 | 0.691 | 0.814 | 0.708 | 0.964 | 0.746 | 0.910 | 0.714 | 1.092 | 0.769 |

*Table 11.* Multivariate time series forecasting results across five datasets. The historical length $HL = 672$ and the forecasting length $FL \in \{96, 192, 336, 720\}$. The best is in **bold** and second-best is underlined.

| Models | | DSENet (Ours) | | TimeFilter (ICML, 2025) | | MSGNet (AAAI, 2024) | | TimeMixer (ICLR, 2024) | | iTransformer (ICLR, 2024) | | MICN (ICLR, 2023) | | TimesNet (ICLR, 2023) | | PatchTST (ICLR, 2023) | | NsT (NeurIPS, 2022) | |
|---|---|---|---|---|---|---|---|---|---|---|---|---|---|---|---|---|---|---|---|
| Metrics | | MSE | MAE | MSE | MAE | MSE | MAE | MSE | MAE | MSE | MAE | MSE | MAE | MSE | MAE | MSE | MAE | MSE | MAE |
| ETTh1 | 96 | **0.387** | **0.418** | 0.476 | 0.460 | 0.421 | 0.449 | 0.517 | 0.498 | 0.388 | 0.420 | 0.761 | 0.651 | 0.450 | 0.463 | 0.414 | 0.420 | 0.787 | 0.653 |
| | 192 | **0.424** | **0.437** | 0.490 | 0.474 | 0.465 | 0.477 | 0.476 | 0.477 | 0.441 | 0.443 | 0.591 | 0.544 | 0.468 | 0.476 | 0.444 | 0.454 | 0.653 | 0.589 |
| | 336 | **0.437** | **0.450** | 0.490 | 0.487 | 0.471 | 0.484 | 0.776 | 0.617 | 0.487 | 0.463 | 0.772 | 0.643 | 0.465 | 0.479 | 0.499 | 0.487 | 0.736 | 0.632 |
| | 720 | **0.481** | **0.482** | 0.639 | 0.581 | 0.541 | 0.531 | 0.555 | 0.532 | 0.547 | 0.534 | 0.934 | 0.729 | 0.521 | 0.541 | 0.500 | 0.531 | 0.632 | 0.583 |
| ETTh2 | 96 | **0.285** | **0.343** | 0.300 | 0.358 | 0.381 | 0.418 | 0.295 | 0.362 | 0.327 | 0.371 | 0.393 | 0.425 | 0.395 | 0.427 | 0.345 | 0.392 | 0.415 | 0.451 |
| | 192 | **0.351** | **0.388** | 0.399 | 0.415 | 0.452 | 0.449 | 0.366 | 0.411 | 0.387 | 0.410 | 0.494 | 0.485 | 0.445 | 0.462 | 0.404 | 0.433 | 0.425 | 0.449 |
| | 336 | **0.342** | **0.394** | 0.465 | 0.456 | 0.429 | 0.447 | 0.698 | 0.589 | 0.433 | 0.445 | 0.772 | 0.643 | 0.452 | 0.465 | 0.449 | 0.458 | 0.434 | 0.455 |
| | 720 | **0.402** | **0.445** | 0.530 | 0.521 | 0.523 | 0.511 | 0.553 | 0.532 | 0.482 | 0.488 | 1.979 | 1.083 | 0.508 | 0.499 | 0.470 | 0.488 | 0.470 | 0.486 |
| ETTm1 | 96 | 0.291 | 0.345 | **0.286** | 0.346 | 0.354 | 0.394 | 0.310 | 0.353 | 0.334 | 0.368 | 0.340 | 0.394 | 0.334 | 0.375 | 0.384 | 0.393 | 0.597 | 0.488 |
| | 192 | **0.325** | 0.373 | 0.347 | 0.375 | 0.381 | 0.414 | 0.355 | 0.381 | 0.377 | 0.391 | 0.401 | 0.431 | 0.407 | 0.413 | 0.393 | 0.408 | 0.490 | 0.467 |
| | 336 | **0.360** | 0.395 | 0.380 | 0.400 | 0.443 | 0.449 | 0.380 | 0.403 | 0.426 | 0.420 | 0.465 | 0.483 | 0.414 | 0.422 | 0.403 | 0.424 | 0.569 | 0.488 |
| | 720 | **0.435** | **0.426** | 0.441 | 0.437 | 0.476 | 0.468 | 0.442 | 0.429 | 0.491 | 0.459 | 0.526 | 0.515 | 0.478 | 0.450 | 0.452 | 0.448 | 0.598 | 0.514 |
| ETTm2 | 96 | **0.163** | 0.253 | 0.170 | 0.259 | 0.206 | 0.291 | 0.194 | 0.288 | 0.179 | 0.290 | 0.227 | 0.322 | 0.196 | 0.285 | 0.178 | 0.270 | 0.316 | 0.374 |
| | 192 | **0.226** | **0.298** | 0.261 | 0.316 | 0.268 | 0.330 | 0.232 | 0.306 | 0.292 | 0.316 | 0.302 | 0.369 | 0.280 | 0.338 | 0.251 | 0.316 | 0.376 | 0.408 |
| | 336 | **0.292** | **0.344** | 0.378 | 0.373 | 0.324 | 0.365 | 0.425 | 0.448 | 0.321 | 0.367 | 0.424 | 0.464 | 0.336 | 0.373 | 0.305 | 0.354 | 0.395 | 0.410 |
| | 720 | **0.385** | **0.399** | 0.412 | 0.409 | 0.417 | 0.422 | 0.405 | 0.420 | 0.412 | 0.409 | 0.655 | 0.591 | 0.423 | 0.421 | 0.407 | 0.420 | 0.480 | 0.463 |
| Exchange | 96 | **0.089** | 0.241 | 0.110 | 0.209 | 0.345 | 0.417 | 0.185 | 0.327 | 0.111 | 0.281 | 1.152 | 0.899 | 0.207 | 0.343 | 0.113 | 0.242 | 0.552 | 0.557 |
| | 192 | 0.234 | 0.379 | **0.216** | **0.327** | 0.730 | 0.591 | 0.465 | 0.505 | 0.259 | 0.382 | 0.263 | 0.404 | 0.489 | 0.530 | 0.221 | 0.344 | 0.813 | 0.685 |
| | 336 | **0.387** | **0.480** | 0.537 | 0.510 | 1.224 | 0.765 | 0.633 | 0.604 | 0.435 | 0.495 | 0.746 | 0.695 | 0.893 | 0.711 | 0.416 | 0.484 | 0.900 | 0.706 |
| | 720 | **0.845** | **0.737** | 0.854 | 0.786 | 1.681 | 0.914 | 1.564 | 0.953 | 1.471 | 0.911 | 1.562 | 0.989 | 1.617 | 0.933 | 1.159 | 0.817 | 2.332 | 1.099 |

## H. Performance in Multiple Real-Word Scenarios

To comprehensively evaluate the generalization ability of DSENet and its performance on other tasks, we conduct experiments on five widely used real-world time-series datasets, including ETTh1, ETTh2, ETTm1, ETTm2, and Exchange. We compare DSENet with both classical baselines and recent state-of-the-art models in the time-series forecasting literature, including PatchTST (Nie, 2022), Nonstationary Transformer(NsT, (Liu et al., 2022)), TimesNet (Wu et al., 2022), MICN (Wang et al., 2023), iTransformer(Liu et al., 2023), TimeMixer (Wang et al., 2024), MSGNet (Cai et al., 2024) and TimeFilter (Hu et al.,

2025). In our experimental setup, a typical historical length $HL = 96$ and a long historical length of $HL = 672$ is adopted to demonstrate the capability of our model in modeling long-range dependencies, while the forecasting horizon is denoted by $FL \in \{96, 192, 336, 720\}$. All models are trained under a unified setting on NVIDIA RTX 4090 GPU at Ubuntu 22.04. The experimental results are reported in Table 10 and Table 11.

From the results in Table 11, our model achieves excellent results across various historical input lengths, demonstrating particularly outstanding performance when handling extremely long historical data. It consistently delivers robust results across multiple datasets. This indicates the model possesses significant advantages in modeling long-range dependencies and preserving information. Simultaneously, the introduction of the dual-stream augmentation mechanism enhances the model's sensitivity to local dynamic changes. For real-world temporal tasks, critical decisions often rely not solely on recent observations but require integrating slowly evolving trends, periodic patterns, and sparse yet crucial anomaly signals from long-term history. Furthermore, owing to its linear computational complexity and low memory consumption, our model demonstrates immense potential in time series forecasting applications.

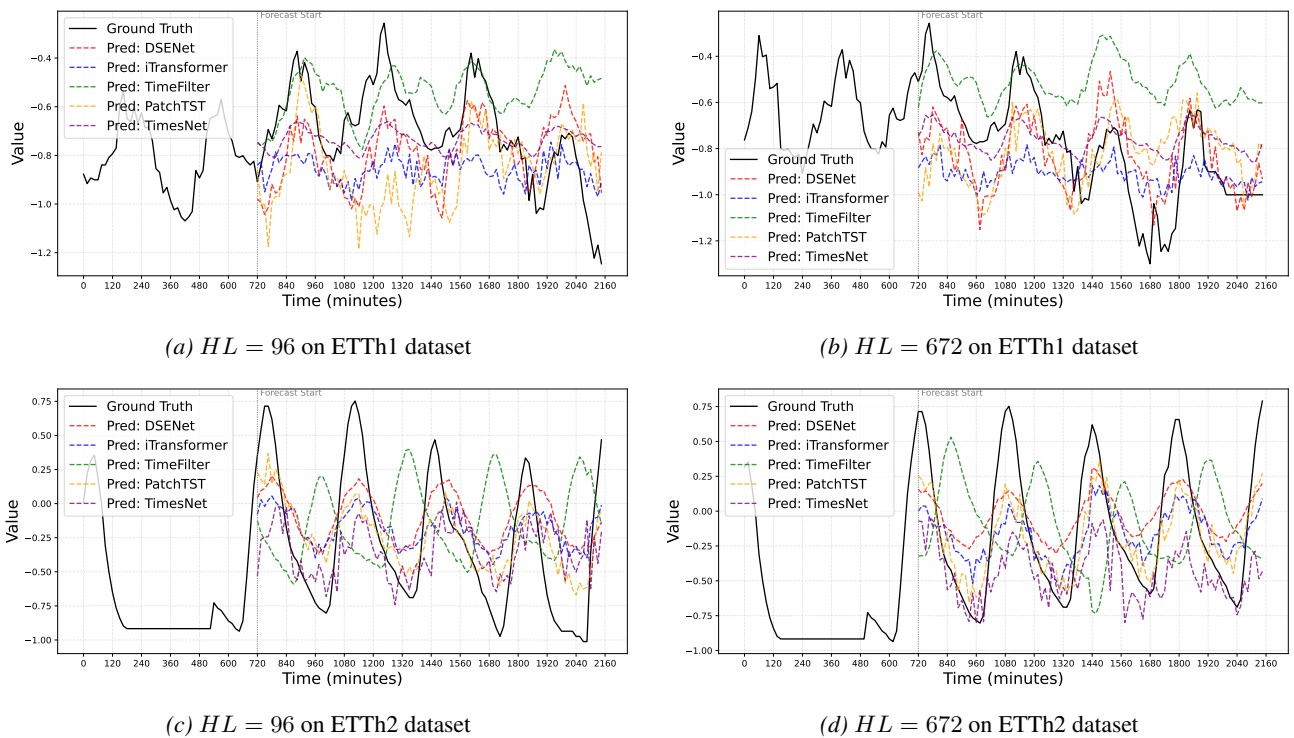

*(a) $HL = 96$ on ETTh1 dataset*      *(b) $HL = 672$ on ETTh1 dataset*

*(c) $HL = 96$ on ETTh2 dataset*      *(d) $HL = 672$ on ETTh2 dataset*

*Figure 16.* Visualization results of ETTh1 and ETTh2 datasets of DSENet and other four outstanding models. (a), (c) are under $HL = 96$ and (b), (d) are under $HL = 672$.

As shown in Figure 16, the visualization of actual and predicted values demonstrates our model's prediction results on the real dataset. Using the ETTh1 and ETTh2 datasets as examples, we select four other outstanding models: TimeFilter, iTransformer, PatchTST, and TimesNet for visualization comparison. The results demonstrate that our model exhibits the most outstanding performance, maximally capturing both the global trends and local variations of the GT curves.

