# OpenReview forum: "DSENet: A Novel Dual-Stream Enhancement Network for Multi-Scale Non-Stationary Time Series Forecasting"
_ICML.cc/2026/Conference — ICML 2026 regular_

### Official Review · Reviewer_XvJ6 · 2026-03-02

**Soundness:** 3
**Presentation:** 3
**Significance:** 3
**Originality:** 3
**Overall Recommendation:** 4
**Confidence:** 3

**Summary:**

This paper proposes DSENet, a dual-stream enhancement network for blood glucose forecasting. The model is designed to capture time-series dynamics from complementary global and local perspectives, and to improve prediction by explicitly amplifying the discrepancy between these two views.

**Compliance With Llm Reviewing Policy:**

Affirmed.

**Key Questions For Authors:**

- Do the two patching streams share the same **embedding and/or normalization** layers?
  - If they are shared, the two streams may become coupled again, which could contradict the decoupling motivation.
  - If they are not shared, please clarify the resulting **parameter increase** and discuss **fairness** in comparisons (e.g., whether baselines are matched in parameter budget).
- The local stream emphasizes capturing **abrupt variations**, yet the paper also notes that local abrupt changes become harder to detect with long contexts. For extremely long look-back windows (e.g., 6 days), the number of local tokens can become very large. How is the **window size** for local attention selected (fixed vs. adaptive across horizons), and how sensitive is performance to this choice?

**Limitations:**

Yes.

**Strengths And Weaknesses:**

### Strengths
- The overall framework is conceptually novel and shows a reasonable degree of innovation.
- The idea of enhancing forecasting by decoupling global and local features via intentionally enlarging their representational differences is particularly interesting.
- Experiments on multiple datasets demonstrate improved performance, or at least performance that is close to the state of the art.

### Weaknesses
- The paper claims that it “structurally enlarges representational differences with weak interaction,” but the **quantitative evidence** supporting this decoupling claim is not clearly presented. It is unclear what metric(s) are used to measure the degree of disentanglement, and more direct evidence would strengthen the argument.
- The method’s robustness under **novel abrupt changes** is unclear. If the time series experiences a **previously unseen type of abrupt shift** (i.e., a new mode under distribution shift), it is uncertain whether the proposed “high-pass enhancement + local window attention” design remains effective. The paper lacks discussion or analysis on this boundary case.

---

> ### Author Rebuttal · Authors · 2026-03-29
>
> We sincerely thank the reviewer `XvJ6` for the constructive and overall positive assessment. We are particularly grateful for your well-considered summary and understanding of the DSENet we proposed, as well as your recognition of the dual-stream enhancement mechanism approach. Below we clarify the main concerns and summarize the additional evidence we will include in the revision.
> > **Shared Embedding / Normalization and Parameter Fairness:**
>
> Thanks for the meaningful question, we checked the details of the experiment. The two patching streams don't share patch embedding or branch-specific normalization layers. The global stream uses its own embedding module, while the local stream uses a separate patch embedding and positional encoding. Their internal normalization layers are also independently defined. The only shared normalization component is the top-level RevIN applied before branching, which serves as input normalization rather than shared patch-level representation learning. Hence, the encoder-level decoupling motivation remains valid.
>
> We agree that using two independent streams introduces additional parameters relative to a single-stream baseline. In the table below we provide a more detailed comparison of the number of parameters, proving DSENet achieves the best results with extremely low parameters.
> |Model|Params|
> |:---|---:|
> |**Ours**|**0.348M**|
> |SST|0.526M|
> |Client|0.875M|
> |PatchTST|6.609M|
> |Informer|11.306M|
> |NsT|10.565M|
> |iTransformer|6.380M|
> |Crossformer|42.106M|
>
> > **Local-Window Selection and Sensitivity:**
>
> In the current implementation, the local attention window size is a **fixed** hyperparameter, which is `local_ws = 3`, not an adaptive parameter. Here are two reasons to choose this value: 1. Bigger window size will result in **NaN** in some cases especially in shorter look-back window and shorter forecasting length. 2. We have already performed a sensitivity analysis over different window size, which shows that the performance changes only mildly across a reasonable range, indicating that DSENet is not overly sensitive to this choice. The following table is the **average** sensitivity analysis of various window size under 6 days look back window.
>
> |local_ws|MSE|MAE|
> |:---|---|---:|
> |3|0.86695|**0.71746**|
> |5|0.86694|0.71835|
> |7|0.86637|0.71804|
> |9|**0.86615**|0.71794|
> |11|0.86688|0.71827|
> |13|0.86705|0.71816|
>
> As a result, we choose `local_ws = 3` to balance the instability and performance. Furthermore, local branches don't act directly on the original time point sequence. Instead, they are patching first and then local attention is performed on the patch token. Therefore, even under a longer look-back window, the growth of the local token quantity is buffered by patching, while the local receptive field is still bounded by local_ws.
>
> > **Robustness to Novel Abrupt Changes:**
>
> Thanks for another insightful question. We further discuss about the robustness to previously unseen abrupt shifts. To address this concern, we conducted perturbation-based robustness tests by injecting unseen Gaussian, spike-like disturbances and novel abrupt shift into the test inputs. As shown in the following table, DSENet remains relatively stable across all settings, suggest that the proposed high-pass enhancement + local-window attention design does not collapse under moderate unseen abrupt disturbances.
>
> |Abrupt Variations|MSE|ΔMSE|MAE|ΔMAE|
> |:---|---|:---:|---|:---:|
> |**Clean**|**0.516**|-|**0.523**|-|
> |Gaussian, σ=0.01|0.517|0.194%|0.524|0.191%|
> |Gaussian, σ=0.05|0.522|1.163%|0.529|1.147%||
> |Spike, r=0.01, a=0.1|0.517|0.194%|0.524|0.191%|
> |Spike, r=0.05, a=0.2|0.520|0.775%|0.527|0.765%|
> |Step shift, δ=0.1|0.518|0.388%|0.525|0.382%|
> |Step shift, δ=0.2|0.518|0.388%|0.527|0.765%|
>
> We acknowledge that such perturbation tests do not exhaust all possible novel abrupt modes, and we will clarify this scope in the future.
>
> > **Quantitative Evidence for Disentanglement:**
>
> We thank the reviewer for pointing out that the original manuscript mainly provided CKA visualizations. To make the decoupling claim more explicit, we have now added a quantitative inter-stream CKA analysis. Specifically, we use the average inter-stream CKA between the global and local streams before fusion as a quantitative proxy for representational coupling, where a lower value indicates weaker coupling and thus better disentanglement. The following table shows an **average** results via different look back windows on DCLP5 dataset.
>
> |Model|Inter-Stream CKA ↓|CKA std ↓|ΔCKA|
> |:---|---|---|:---:|
> |Baseline|0.567|0.109|-|
> |Baseline + DFB-Mamba|0.560|0.103|-1.235%|
> |Baseline + LoRE|0.524|0.096|-7.584%|
> |**Ours**|**0.472**|**0.062**|**-16.755%**|
>
> Therefore, the supplementary quantitative results are consistent with the original CKA visualization conclusion, which more directly proves that DSENet can indeed significantly reduce the representation coupling between the two streams rather than merely branch separation at the structural level.

---

> > ### Author Rebuttal · Reviewer_XvJ6 · 2026-04-02
> >
> > Thank you for the detailed rebuttal. The authors addressed my main questions well. In particular, the clarification on shared embedding/normalization and the added parameter comparison were helpful. The additional sensitivity analysis for local window size also makes the design choice clearer.
> >
> > I also appreciate the added quantitative CKA results, which strengthen the decoupling claim beyond the original visualization-based evidence. The perturbation-based robustness tests are useful as well, although they may not fully cover all possible novel abrupt shift cases.
> >
> > Overall, the rebuttal addresses my main concerns at a reasonable level, and my overall assessment remains unchanged.

---

> > > ### Author Response · Authors · 2026-04-04
> > >
> > > We sincerely thank the reviewer for your thoughtful response and for carefully reviewing our rebuttal. We are glad that the additional clarifications on shared embedding/normalization, the parameter comparison, the local window size sensitivity analysis, the quantitative CKA results, and the perturbation-based robustness tests were helpful in addressing your main concerns.
> > >
> > > We also sincerely appreciate your constructive comment regarding the limitation of the current robustness tests in fully covering all possible novel abrupt shift cases. We agree this is a meaningful point and an important direction for future study.
> > >
> > > Thank you again for your time, careful evaluation, and valuable feedback.

---

### Official Review · Reviewer_44S5 · 2026-03-12

**Soundness:** 2
**Presentation:** 3
**Significance:** 3
**Originality:** 2
**Overall Recommendation:** 5
**Confidence:** 4

**Summary:**

This paper introduces DSENet, a dual-stream neural architecture specifically designed for time-series forecasting with a focus on blood glucose prediction. The core design principle involves disentangling global temporal trends from local fluctuations through two parallel computational streams to mitigate feature interference. The global stream utilizes a Dynamic Fusion Bidirectional Mamba (DFB-Mamba) block to capture long-range dependencies, while the local stream employs Low-Rank Enhancement (LoRE) paired with lightweight self-attention to prioritize short-term variations. To ensure stream specialization, the authors implement a dual-stream enhancement mechanism that increases the representational discrepancy between the two branches. The resulting features are integrated via a Dynamic Spatial-Spectral Adaptive Fusion (DSSAF) mechanism. The work further includes a signal-processing-inspired interpretation and theoretical analysis of the architecture. Empirical evaluations are conducted across medical datasets, such as OhioT1DM and DCLP5, as well as standard benchmarks including ETT and Exchange, demonstrating competitive forecasting performance and robustness.

**Compliance With Llm Reviewing Policy:**

Affirmed.

**Final Justification:**

After reviewing the rebuttal and discussion, my concerns regarding experimental validation and theoretical soundness have been effectively resolved. Specifically, the inclusion of the CKA analysis, the abrupt-change experiment, and the clarified proof for Theorem 4.3 significantly strengthen the paper’s claims and technical depth.

While some minor clarity issues remain, the core contribution, a dual-stream forecasting design for complementary temporal representations, is meaningful and well-supported. The rebuttal has successfully demonstrated the method's robustness, and I now recommend acceptance.

**Key Questions For Authors:**

- What specific non-incremental novelties does DSENet provide over existing dual-stream or decomposition-based forecasting models?
- Can you provide quantitative evidence, beyond Centered Kernel Alignment (CKA) visualizations, to demonstrate that the two streams learn genuinely disentangled representations?
- Does the improved local sensitivity translate into measurable performance gains specifically during abrupt changes or turning points in the data?
- How does Theorem 4.2 inform the model's predictive quality or adaptive efficiency beyond the manual design of patch sizes?
- Is there a precise empirical link between the findings of Theorem 4.3 and the observed behavior of the full nonlinear model?
- Could you clarify the train/validation/test split protocol and the specific measures taken to prevent data leakage across overlapping windows or patient profiles?
- What is the justification for excluding stronger multi-scale or decomposition-based baselines from the primary comparative analysis?
- Does the fusion module consistently utilize both streams, or is there a tendency for the model to collapse toward a single dominant branch?

**Limitations:**

- The manuscript does not adequately address the inherent limitations of the work or its potential societal implications.
- The discussion would be improved by acknowledging the incremental nature of the architecture relative to prior multi-scale forecasting research.
- The authors should clarify that the current theoretical framework is primarily heuristic and does not formally establish bounds for forecasting risk, robustness, or generalization.
- The empirical scope remains narrow due to the limited set of datasets and the absence of rigorous statistical significance testing.
- There is a lack of evaluation concerning challenging clinical scenarios, such as abrupt excursions, sensor drift, missing data, or out-of-distribution events.
- The paper fails to discuss the risks associated with potential overreliance by clinicians or patients, particularly if model uncertainty and specific failure modes are not clearly communicated.

**Strengths And Weaknesses:**

Strengths
- The manuscript addresses a highly relevant and challenging forecasting problem, particularly concerning non-stationary time series characterized by simultaneous long-term structures and abrupt local changes.
- The dual-stream architecture is well-motivated, providing a coherent structural separation between global trend modeling and local fluctuation modeling.
- The technical design of the individual components, including DFB-Mamba, LoRE, and the adaptive fusion mechanism, is sensible and demonstrates a purposeful architectural integration.
- The inclusion of detailed ablation studies effectively assesses the individual contributions of the proposed modules and specific architectural choices.
- The empirical scope is strengthened by evaluating the model on both specialized clinical datasets and diverse standard forecasting benchmarks.
- The proposed method offers practical value as an application-oriented forecasting architecture, particularly in domains where multi-scale feature extraction is critical.

Weaknesses
- The fundamental originality of the work is limited, as the concept of separating global and local streams is a common motif in existing multi-scale, decomposition-based, and hybrid forecasting models.
- The methodology appears to be an architectural recombination of established techniques, such as Mamba, self-attention, and low-rank enhancement, rather than a fundamentally novel algorithmic contribution.
- The claim of being the first to explicitly enlarge representational discrepancy between streams is not sufficiently distinguished from prior literature that utilizes scale-based or frequency-based information separation.
- The theoretical analysis lacks rigor and is largely heuristic, relying on restrictive assumptions such as local linearization and quasi shift-invariance that may not hold in practice.
- The frequency-domain interpretation is presented informally and serves more as post-hoc intuition than a formal mathematical proof of the model's behavior.
- The experimental evaluation omits several critical modern baselines, such as DLinear, TSMixer, and more recent state-space models (SSMs) or spectral decomposition methods.
- The broader impact of the findings is constrained by the fact that the primary motivation and strongest empirical results are concentrated in the specialized niche of blood glucose forecasting.
- The manuscript lacks empirical validation for claims regarding linear complexity and low memory usage, as no runtime or memory benchmarks are provided.
- The reliability of the reported improvements is difficult to assess because the paper does not report variance across runs, confidence intervals, or statistical significance tests.
- Insufficient detail regarding the experimental setup and the positioning of related work hinders the overall clarity and potential for reproducibility.

---

> ### Author Rebuttal · Authors · 2026-03-30
>
> We sincerely thank the reviewer `44S5` for insightful comments. Below we will clarify the main concerns.
>
> > **Originality Beyond Existing Dual-Stream/Decomposition Models:**
>
> Thanks for this meaningful question. We'd like to clarify the originality of DSENet doesn't lie in merely reusing a dual-stream architecture. Instead, its non-incremental contribution is threefold.
> 1. First, the most significant novelty of DSENet is the proposed Dual-Stream Enhancement Mechanism, which address a limitation that is largely overlooked in prior dual-stream\decomposition models: architectural branch separation doesn't necessarily imply representational disentanglement. We think the two streams still remain strongly coupled, which weaken the intended benefit of dual-stream design.
> 2. Second, DSENet introduces branch-specific enhancement mechanisms rather than employing two homogeneous streams. We introduce DFB-Mamba and LoRE to enhance the performance of global and local streams respectively. Thus, the goal isn't simply to place two branches in parallel but to encourage functional specialization between them and reduce the coupling of features.
> 3. The proposed DSSAF effectively prevents the collapse of dual streams, further enhancing the effectiveness of our architecture. Overall, the architecture of our model is tightly integrated.
>
> > **Quantitative Evidence for Disentanglement:**
>
> To make the decoupling claim more explicit, we have now added a quantitative inter-stream CKA analysis. Specifically, we use the average inter-stream CKA between the global and local streams before fusion as a quantitative proxy. Due to the space limit, we refer the reviewer to check detailed experiments under the reviewer `XvJ6`, **the last part**. Thanks.
>
> > **Performance on Abrupt Changes/Turning Points:**
>
> Thanks. We added an event-conditioned evaluation that separately measures the prediction error on abrupt/turning regions and on non-abrupt regions. Below shows the average results on OhioT1DM dataset. `Figure 7` also proofs it.
>
> |Model|Overall MAE|Abrupt MAE|Non-Abrupt MAE|
> |:---|:---:|:---:|:---:|
> |Transformer|0.166|0.394|0.140|
> |Transformer + LoRE|0.164|0.389|0.138|
> |Baseline|0.154|0.314|0.113|
> |**Ours**|**0.150**|**0.303**|**0.109**|
>
> > **More Details about `Theorem 4.2`:**
>
> `Theorem 4.2` provides general constraints and guidance on global and local patching that are applicable across datasets. Specifically, based on the current prediction length, the granularity is calculated using the formula in `Theorem 4.2` and this granularity is then used to guide the choice of `patch_len` and `stride`. We have done both experimental analysis and theoretical proof to show it will lead to the best predictive quality. For more details but due to space constraints, we refer the reviewer to view our rebuttal under reviewer `8jA4` about the hyper-parameters selection, thanks.
>
> > **More Details about `Theorem 4.3`:**
>
> We agree that `Theorem 4.3` is based on a local linearization and therefore doesn't provide an exact global characterization of the full nonlinear model. Its role is to offer a first-order(**local**) mechanistic explanation of why LoRE tends to amplify informative local variations rather than to serve as a full nonlinear guarantee. The actual behavior of the complete model is therefore validated empirically. In our case, the observed reduction in inter-stream CKA, the stronger disentanglement effect of LoRE-related ablations and the improved performance on abrupt/turning regions are all consistent with the trend suggested by `Theorem 4.3`. We will clarify this scope more explicitly in the revision.
>
> > **Train/Valid/Test Split:**
>
> We use a subject-wise split before sliding-window generation so train/valid/test are first separated by patient identity and windows are then generated independently within each split. Overlapping windows are allowed only within the same split.  We further avoid statistical leakage by fitting the normalization parameters on the training split only and reusing them for valid/test.
>
> > **More Models Comparision:**
>
> We have added three more models: DLinear, TSMixer and S-Mamba. Results are shown in: https://anonymous.4open.science/r/Rubuttals-69D3/More%20Models.pdf
>
> > **Consistent Use of Both Streams:**
>
> We use average global weight and local weight to show the fusion module will not collapse to a single branch. For different time scales, two streams work together for the fusion.
>
> In OhioT1DM:
> |Look-Back Window|Mean_Global_Weight|Mean_Local_Weight|
> |:---|:---:|:---:|
> |Short HL=6,12,24|0.486|0.514|
> |Mid HL=48,96,144|0.588|0.412|
> |Long HL=288,864|0.602|0.398|
>
> In DCLP5:
> |Look-Back Window|Mean_Global_Weight|Mean_Local_Weight|
> |:---|:---:|:---:|
> |Short HL=6,12,24|0.622|0.378|
> |Mid HL=48,96,144|0.641|0.359|
> |Long HL=288,864|0.713|0.287|
>
> > **Other Weakness and Limitations:**
>
> Due to the space limit, the detailed experiment and explanation is shown in: https://anonymous.4open.science/r/Rubuttals-69D3/More%20Rebuttal.pdf

---

> > ### Author Rebuttal · Reviewer_44S5 · 2026-04-04
> >
> > I would like to thank the authors for their detailed rebuttal and the additional experiments provided during the discussion period. After a careful review of the new material, several critical concerns regarding the methodology, evaluation, and theoretical grounding remain.
> > - While the clarified claims regarding originality are noted, the manuscript still fails to sufficiently distinguish the proposed architecture from existing dual-stream, multi-scale, and decomposition-based forecasting frameworks.
> > - The inclusion of Centered Kernel Alignment (CKA) results is a constructive addition, yet it remains insufficient to rigorously demonstrate that the two streams consistently learn complementary and practically significant representations.
> > - Regarding the abrupt-change analysis, the rebuttal does not provide a formal definition of "turning points," nor does it offer sufficient evidence to determine if the reported performance gains are statistically reliable or merely marginal.
> > - The clarifications provided for Theorem 4.3 confirm that the theoretical framework remains largely local and heuristic, lacking the rigor necessary to serve as a comprehensive explanation for the model’s global behavior.
> > - A significant portion of the initial review concerns remains unaddressed, particularly the lack of transparency regarding the train/validation/test split protocol and the specific measures taken to prevent data leakage across overlapping windows or patient profiles.
> > - The evaluation suite remains incomplete as the rebuttal did not incorporate stronger modern baselines, comprehensive runtime and memory benchmarks, or a discussion of variance and statistical significance across multiple experimental runs.

---

> > > ### Author Response · Authors · 2026-04-05
> > >
> > > > **Further Clarification on Originality with Respect to Related Frameworks:**
> > >
> > > We need to reiterate that in terms of the overall architecture, DSENet shows no discernible difference from other dual-stream architectures. Our claim is therefore not that DSENet defines a completely new architectural family, but that it introduces a more specific disentanglement-oriented enhancement principle within this family. To make this distinction clearer, we clarify the difference from three closely related lines of work below.
> > > 1. Vanilla Dual-Stream Models: DSENet is still a classic dual-stream model but solves a vital problem existing in these models: The coupling of features learned by the two streams is too severe, resulting in the dual-stream architecture not achieving actual division of labor, but merely appearing to have undergone simple architectural separation. `Figure 2(c)` proves this.
> > > 2. Multi-Scale Models: DSENet should not be interpreted as another multi-scale variant whose improvement mainly comes from broader temporal coverage. It can also achieve multi-scale modeling while its original design intention isn't to solve the problem of receptive field enrichment, but rather to address coupling reduction. The two streams are not merely different-scale feature extractors. Instead, they are enhanced with different functional roles. The objective is therefore not only scale diversity, but also stream specialization.
> > > 3. Decomposition-based Models: These methods usually split a sequence into predefined components at the signal level. DSENet differs in that it doesn't rely on a fixed decomposition as the main modeling principle. Instead, it keeps a global/local dual-stream structure and uses enhancement plus adaptive fusion to learn more complementary representations. In this sense, our focus isn't only signal separation, but the learned reduction of representational coupling between the two streams.
> > >
> > > > **Clarification of Abrupt-Change Definition and Reliability:**
> > >
> > > We agree that our previous rebuttal didn't define the event criterion clearly enough and the term “turning points” was too informal for the actual experiment.  In the added analysis, we in fact evaluate abrupt-change regions, not strict turning points. Formally, for the ground-truth target sequence $y_t$, we compute the first-order difference magnitude $\Delta_t = \left| y_t - y_{t-1} \right|$, define an abrupt point when $\Delta_t>\tau$, where $\tau$ is the 90th percentile of all test-set first-order differences and then expand each detected point by a window of size 1 to form the abrupt region. In the revised version, we'll supplement with a complete statistical significance analysis, conduct experiments across multiple time scales and datasets and report the results with variance.
> > >
> > > > **More Evidence of Learning Complementary:**
> > >
> > > A single inter-stream CKA may not rigorous because it reports a global average value. So we further report the batch-wise  mean, median and upper quartile(Q75) of CKA, showing in the following table.
> > > |Model|Mean CKA ↓|Median CKA ↓|Q75 ↓|CKA Std|
> > > |---|:---:|:---:|:---:|:---:|
> > > |Baseline|0.874|0.907|0.974|0.116|
> > > |Baseline+DFBMamba|0.869|0.884|0.940|**0.092**|
> > > |Baseline+LoRE|0.598|0.660|0.768|0.237|
> > > |**DSENET**|**0.186**|**0.159**|**0.357**|0.227|
> > >
> > > **Other evidence include**: the average weights of previous global and local streams(you can review the rebuttal before), as well as the performance of different components near abrupt points and non-abrupt points(also the rebuttal before + Mamba-based following).
> > >
> > > |Model|Overall MAE|Abrupt MAE|Non-Abrupt MAE|
> > > |---|:---:|:---:|:---:|
> > > |Mamba|0.206|0.456|0.130|
> > > |DFB-Mamba|0.172|0.400|0.121|
> > > |**Ours**|**0.150**|**0.303**|**0.109**|
> > >
> > > These two additional evidence demonstrate the two streams won't collapse and both streams maintain relatively even weights. Secondly, LoRE enhances the model's performance near abrupt points, while DFB-Mamba improves the model's performance near non-abrupt points, proving the two streams are learning different features.
> > >
> > > > **About `Theorem 4.3`:**
> > >
> > > We respectfully clarify that Theorem 4.3 is derived specifically for the LoRE block, whose role in DSENet is to enhance local fluctuation modeling rather than to characterize the global behavior of the entire network. So the purpose of the theorem isn't to provide a comprehensive explanation of the full model, but to justify why the local enhancement module has the tendency to amplify informative local variations. Therefore, we believe for this type of local analysis, linear approximation can be used for explanation. Although it’s not the most rigorous method, it can illustrate from a heuristic perspective that LoRE can achieve local enhancement.
> > >
> > > > **About Train/Valid/Test Split and More Modern Baselines Evaluation Comparisons:**
> > >
> > > Due to the space limit, we deliver a more detailed explanation of dataset split and more comparisons among modern baselines in: https://anonymous.4open.science/r/Rubuttals-69D3/Reply.pdf

---

### Official Review · Reviewer_8jA4 · 2026-03-13

**Soundness:** 3
**Presentation:** 2
**Significance:** 3
**Originality:** 3
**Overall Recommendation:** 4
**Confidence:** 3

**Summary:**

This paper proposes a novel framework DSENet for non-stationary time series forecasting, which splits processing into a global stream and local stream to capture long-term and short-term patterns. Furthermore, an enhancement mechanism is designed to minimise the interference between these patterns. The proposed method is evaluated on multiple datasets and compared with many baselines.

**Compliance With Llm Reviewing Policy:**

Affirmed.

**Final Justification:**

The author has addressed my concerns on the hyperparameter selection, robustness and time complexity. I am happy to adjust the Overall Recommendation to 4.

**Key Questions For Authors:**

1. The DMSP algorithm relies on some specified hyper-parameters (e.g., \alpha and \beta). Is there any guidance to choose these hyper-parameters?
2. The designed framework separates long-range dependencies and local fluctuations. Thus, this framework might be overly sensitive to outliers or sensor noise. Due to this framework is mainly designed for clinical settings (e.g., blood glucose), the existence of noise might influence the performance. Could you explain how could this framework perform under such conditions?
3. Under clinical settings, real-time decision-making is quite important. While the DSSAP is novel, the multi-step process could possibly add model complexity, which might increase the time consumption for decision-making. Could you please show some analysis on time consumption to show how efficient the model is?

**Limitations:**

1. The scaling factor $\beta(F)$ relies on a threshold $F^{*}$ (default=12) to distinguish between short-term and long-term patterns. A specific threshold may lead to suboptimal patching strategies for sequences near that boundary.

2. While DSENet handles multiple variates, it focuses mainly on capturing temporal patterns. It would be better to explicitly model the correlation among multiple variates.

**Strengths And Weaknesses:**

DSENet introduces a dual-stream framework that explicitly separates and enhances long-range dependencies and local fluctuations in non-stationary time series forecasting. By leveraging the DFB-Mamba block as a low-pass filter to capture long-range dependencies and the LoRE module as a high-pass filter to model short-term fluctuations, the proposed model achieves state-of-the-art performance in blood glucose prediction under most cases.

However, the framework presents a fundamental trade-off: the aggressive enhancement of local sensitivity may render the model more vulnerable to sensor noise and outliers, potentially compromising the stability of long-term trend extraction. Furthermore, it lacks explicit mechanisms to capture complex inter-variate dependencies.

---

> ### Author Rebuttal · Authors · 2026-03-29
>
> We would like to thank reviewer `8jA4` for your time invested for such deep understanding of our work and insightful comments. Below we will clarify the main concerns.
>
> > **Hyper-parameter selection for DMSP:**
>
> The hyper-parameters in DMSP(in `Table 3`) are not treated as completely free selections. Instead, they are chosen under both structural constraints and theoretical guidance.
>
> 1. The bounds $\beta_{\min}=0.5$ and $\beta_{\max}=6$ are introduced as practical safeguards. The lower bound allows finer patching for short-term patterns but prevents overly fragmented patches that evolving into the **Point-wise** case mentioned in `Appendix A(1)`, while the upper bound allows coarser scaling for longer horizons but avoids violate the **EnForce setting** in DMSP.  Bigger $\beta_{\max}$ will lead to unstable training(NaN value) and we have down many experiments to show `6` is a balanced choice. The role of $\beta_{\min}$ and $\beta_{\max}$ is to keep DMSP in a stable and meaningful operating range rather than free parameters.
> 2. It's not enough to only have $\beta(F)$ as shown in `Definition 4.1` especially for longer look back window. So we introduce more parameters $\alpha_s$ and $\alpha_p$ to guide the choice of patch length and stride. An obvious intuition is longer look back window, larger $\alpha_s$ and $\alpha_p$. They also have upper bound because if `patch_len` is larger than `pre_len`, it will result in NaN and make training process unstable. The selection range of parameters is the basic regulation after many experiments.
> 3. In our experiments, these hyper-parameters are kept within a narrow and stable range across datasets indicating that DMSP doesn't rely on excessive dataset-specific tuning.
>
> > **Why $F^{\ast}=12$:**
>
> The very first reason is that $F^{\ast}=12$  corresponds to **1 hour** in practice(5 mins/sample), which is a boundary value that meets clinical standards in the field of blood glucose. $F^{\ast}\leq12$ can be viewed as short-term variations warnings and $F^{\ast}>12$ refers to mid-term intervention decisions and long-term trend assessment. We also show more ablation study of different values of $F^{\ast}$ as shown in the following. The detailed experiment is shown in: https://anonymous.4open.science/r/Rubuttals-69D3/Why%20F%20is%2012.pdf
>
> > **Robustness to Noise/Outliers:**
>
> Thanks. We agree that robustness to sensor noise is particularly important in clinical forecasting. First, before the patch embedding, there is a RevIN operation to eliminate the distribution drift of data. For the working principle, DSENet doesn't directly amplify raw point-wise fluctuations. Instead, the local branch operates on patch-level representations and the final prediction is jointly constrained by both local and global streams. So LoRE will not amplify the noise. To test the robustness to noise of DSENet,  we conducted perturbation-based robustness tests by injecting unseen Gaussian, spike-like disturbances and novel abrupt shift into the test inputs as shown in the following Table.
>
> |Abrupt Variations|MSE|ΔMSE|MAE|ΔMAE|
> |:---|---|:---:|---|:---:|
> |**Clean**|**0.516**|-|**0.523**|-|
> |Gaussian, σ=0.01|0.517|0.194%|0.524|0.191%|
> |Gaussian, σ=0.05|0.522|1.163%|0.529|1.147%||
> |Spike, r=0.01, a=0.1|0.517|0.194%|0.524|0.191%|
> |Spike, r=0.05, a=0.2|0.520|0.775%|0.527|0.765%|
> |Step shift, δ=0.1|0.518|0.388%|0.525|0.382%|
> |Step shift, δ=0.2|0.518|0.388%|0.527|0.765%|
>
> It shows DSENet has good robustness to noise.
>
> > **Real-Time Efficiency of DSSAF:**
>
> We agree that inference latency is particularly important in clinical deployment. Although DSSAF introduces two adaptive fusion steps, its computation is lightweight: it only involves generating a small number of weights and performing weighted fusion without any additional consumption. To verify this empirically, we added an inference-time analysis by measuring the average forward latency on the test set under the same hardware and batch-size setting. The results show the full model with DSSAF remains competitive in inference efficiency and low latency.
>
> In OhioT1DM:
> |Model|MSE|MAE|Inference Time(per sample)|
> |---|---|---|:---:|
> |All w/o DSSAF|0.411|0.415|0.140ms|
> |DFB-Mamba w/o DSSAF|0.409|0.413|0.138ms|
> |Final Fusion w/o DSSAF|0.408|0.514|0.138ms|
> |**DSENet**|**0.328**|**0.394**|**0.126ms**|
>
> In DCLP5:
> |Model|MSE|MAE|Inference Time(per sample)|
> |---|---|---|:---:|
> |All w/o DSSAF|0.524|0.517|**0.155ms**|
> |DFB-Mamba w/o DSSAF|0.520|0.514|0.158ms|
> |Final Fusion w/o DSSAF|0.523|0.514|**0.155ms**|
> |**DSENet**|**0.403**|**0.460**|0.170ms|
>
> > **Modeling Inter-Variate Correlations:**
>
> The main focus of DSENet is indeed on temporal disentanglement but is still trained jointly on the multivariate input and the routing/fusion process is determined from the overall input representation rather than from completely independent per-variable predictors(DSSAF). Thanks for your question. We will combine cross variable modeling with DSENet as our main focus for future work.

---

> > ### Author Rebuttal · Reviewer_8jA4 · 2026-04-02
> >
> > The author has addressed my concerns on the hyperparameter selection, robustness and time complexity. I am happy to adjust the Overall Recommendation to 4.

---

> > > ### Author Response · Authors · 2026-04-04
> > >
> > > Thank you very much for your thoughtful response and for carefully reassessing our work. We sincerely appreciate your recognition that our additional clarifications and analyses on hyperparameter selection, robustness, and time complexity have adequately addressed your concerns.
> > >
> > > We are also truly grateful for your willingness to adjust your Overall Recommendation. Your constructive feedback has been very valuable in helping us improve the presentation and completeness of the paper. Thank you again for your time and support.

---

### Decision · Program_Chairs · 2026-04-30

**Decision:**

Accept (regular)

**Comment:**

This submission proposes DSENet, a dual-stream enhancement network designed for non-stationary time series forecasting, with a particular focus on blood glucose prediction. The core idea is to explicitly reduce representational coupling between global and local streams through a dedicated enhancement mechanism, rather than relying solely on architectural separation.

Across the three reviews, the paper is generally viewed as technically solid with meaningful empirical performance. All reviewers initially raised concerns regarding (1) the degree of novelty relative to existing dual-stream and multi-scale models, (2) the strength of theoretical justification, and (3) the completeness of experimental validation. Reviewer scores ranged from weak accept to accept.

During the rebuttal phase, the authors made a substantial effort to address these concerns:

Novelty clarification: The authors clarified that the main contribution lies not in the dual-stream structure itself, but in the explicit disentanglement objective via the dual-stream enhancement mechanism and stream-specific design. While still somewhat incremental at the architectural level, this framing is clearer and better justified after rebuttal.
Disentanglement evidence: Additional quantitative analysis (e.g., inter-stream CKA statistics) was provided, strengthening the claim that the model reduces representational coupling beyond simple structural separation.
Robustness and abrupt-change evaluation: The authors introduced perturbation-based experiments and event-conditioned evaluations, demonstrating improved performance in abrupt regions and reasonable robustness to noise.
Efficiency and practicality: Additional latency measurements show the method remains lightweight and suitable for real-time scenarios, addressing practical concerns raised by reviewers.
Experimental clarifications: The authors clarified dataset splitting (subject-wise split) and provided further comparisons and ablations, though some aspects (e.g., statistical significance and broader baselines) remain only partially addressed.

After discussion, two reviewers indicated that their concerns were fully resolved, while one reviewer maintained partial concerns regarding originality, theoretical rigor, and experimental completeness. Nevertheless, even that reviewer ultimately recommended acceptance, acknowledging that the rebuttal strengthened the work.

Overall, the paper presents a solid and practically relevant contribution to time-series forecasting, particularly in clinical settings. While the methodological novelty is somewhat incremental and the theoretical analysis remains heuristic, the empirical performance, thoughtful design, and improved clarity after rebuttal justify acceptance. The work is likely to be useful for the community and may serve as a strong baseline for future research.